# Contrasting geochemical and fungal controls on decomposition of lignin and soil carbon at continental scale

Wenjuan Huang [1,7], Wenjuan Yu [1,7] ✉, Bo Yi [1], Erik Raman[2], Jihoon Yang[2], Kenneth E. Hammel [3,4], Vitaliy I. Timokhin[5], Chaoqun Lu[1], Adina Howe[2], Samantha R. Weintraub-Leff [6] & Steven J. Hall [1]

Lignin is an abundant and complex plant polymer that may limit litter decomposition, yet lignin is sometimes a minor constituent of soil organic carbon (SOC). Accounting for diversity in soil characteristics might reconcile this apparent contradiction. Tracking decomposition of a lignin/litter mixture and SOC across different North American mineral soils using lab and field incubations, here we show that cumulative lignin decomposition varies 18-fold among soils and is strongly correlated with bulk litter decomposition, but not SOC decomposition. Climate legacy predicts decomposition in the lab, and impacts of nitrogen availability are minor compared with geochemical and microbial properties. Lignin decomposition increases with some metals and fungal taxa, whereas SOC decomposition decreases with metals and is weakly related with fungi. Decoupling of lignin and SOC decomposition and their contrasting biogeochemical drivers indicate that lignin is not necessarily a bottleneck for SOC decomposition and can explain variable contributions of lignin to SOC among ecosystems.

Lignin is one of the most abundant biopolymers in the terrestrial biosphere and protects other components of plant tissue from microbial attack. Traditionally, it was assumed that lignin limits litter decomposition[1,2] and contributes substantially to soil organic carbon (SOC)[3,4]. More recently, lignin's importance in controlling litter and SOC decomposition has become controversial. Lignin might decompose fastest during early stages of litter decomposition[5,6] and lignin-derived C could be less persistent in soil than other C components[7]. The contradictory views related to lignin decomposition and its contributions to SOC might be related to biogeochemical differences among ecosystems. The persistence of lignin relative to SOC and other litter components may vary systematically with climatic, geochemical, and microbial characteristics across diverse soils[8], but the controls on lignin, litter, and SOC decomposition have rarely been investigated together or across a wide range of climatic, geochemical or microbial variation.

Climate can effectively predict litter decomposition at site to continental scales[2], but climate may affect decomposition of different C forms in different ways. For example, although high temperature and precipitation generally increase litter decomposition[2], they can also increase mineral weathering and C stabilization with reactive metals[9,10], which may specifically bind lignin-derived C[11–13]. In addition to climate, the ratio of lignin to nitrogen (N) is another conventionally important predictor of litter decomposition, but it may also have different relationships with lignin, litter or SOC decomposition. Greater litter N content may increase lignin and litter decomposition by alleviating microbial N limitation[1,14], whereas increased N availability may also decrease decomposition of lignin or SOC by suppressing the

[1]Department of Ecology, Evolution, and Organismal Biology, Iowa State University, Ames, IA, USA. [2]Department of Agricultural and Biosystems Engineering, Iowa State University, Ames, IA, USA. [3]U.S. Forest Products Laboratory, Madison, WI, USA. [4]Department of Bacteriology, University of Wisconsin, Madison, WI, USA. [5]Great Lakes Bioenergy Research Center, University of Wisconsin, Madison, WI, USA. [6]National Ecological Observatory Network, Battelle, Boulder, CO, USA. [7]These authors contributed equally: Wenjuan Huang, Wenjuan Yu. ✉e-mail: yuwingjane@gmail.com

production of oxidative enzymes[4]. Both mechanisms may occur in the same soils depending on the stage of litter decomposition; N may stimulate early stages while inhibiting later stages of litter decay[14,15]. However, the overall importance of N relative to other soil characteristics remains poorly understood.

Soil geochemical characteristics might also be important predictors of lignin and litter decomposition in ways that differ from bulk SOC. Soil minerals and metals can protect SOC from microbial decomposition through sorption, co-precipitation, and polyvalent cation bridging[16]. In some soils, lignin-derived C may preferentially associate with iron (Fe) and aluminum (Al) relative to bulk litter or bulk SOC[11,13,17], and these metals might therefore be more important for limiting the decomposition of lignin vs. other compounds in litter or SOC. Manganese (Mn) might also protect C by physical or chemical mechanisms[18], yet because some forms of Mn are powerful oxidants, increased Mn availability can also stimulate lignin decomposition[19]. Calcium (Ca) might also have a contrasting relationship between the decomposition of different substrates: Ca promotes physicochemical protection of SOC[20], yet Ca availability may stimulate lignin-degrading fungi and litter decomposition[21]. However, the consistency of relationships between metals and decomposition rates of lignin, litter, and/or SOC across diverse ecosystems remains unresolved.

Besides geochemistry, microbial composition, and abundance may also have different impacts on the decomposition of different C forms. Many microbial taxa perform similar functions, and it remains elusive whether microbial composition explains process rates[22]. Yet, due to lignin's complex biochemical structure, only a small subset of microbial taxa (white-, brown-, and soft-rot fungi, and certain bacteria) has been conclusively demonstrated to cleave the lignin macromolecule at the propyl sidechain, which is likely the rate-limiting step in lignin decomposition[23-25]. In contrast, microbial composition may be less important for SOC decomposition because physical restrictions on microbial access to C substrates may predominantly limit SOC mineralization[26].

To test competing viewpoints and potential mechanisms underlying the role of lignin in organic matter decomposition, we measured decomposition of lignin, bulk litter, and SOC via a uniform and quantitative isotopic method from mineral soil samples collected across broad biophysical gradients. We used 20 sites from the US National Ecological Observatory Network (NEON) that span diverse ecosystems and climatic zones (tundra to tropics). These particular samples were not necessarily representative of North American soils as a whole, but were instead selected to span a broad range of biogeochemical properties thought to influence C decomposition and accrual; they included 9 of 12 orders in the USDA soil taxonomy (Fig. 1, Supplementary Fig. 1 and Table 1). Previous examinations of lignin decomposition often relied on indirect methods, such as acid-unhydrolyzable residue to approximate lignin content[27], oxidation of simple substrates as a measure of potential ligninolytic enzymes[4], or use of lignin monomers rather than polymers in incubation experiments[28]. These methods can substantially underestimate or overestimate the lignin content of litter and soil as well as the activities of ligninolytic enzymes[29,30]. However, C isotope-labeled, high-molecular-weight synthetic lignin provides a tracer that allows unambiguous quantitative measurement of lignin decomposition[31].

Here, we combine isotope-labeled lignin with natural abundance litter derived from a C4 grass and add these mixtures to separate soil samples for incubation in the lab, enabling us to quantify lignin, litter,

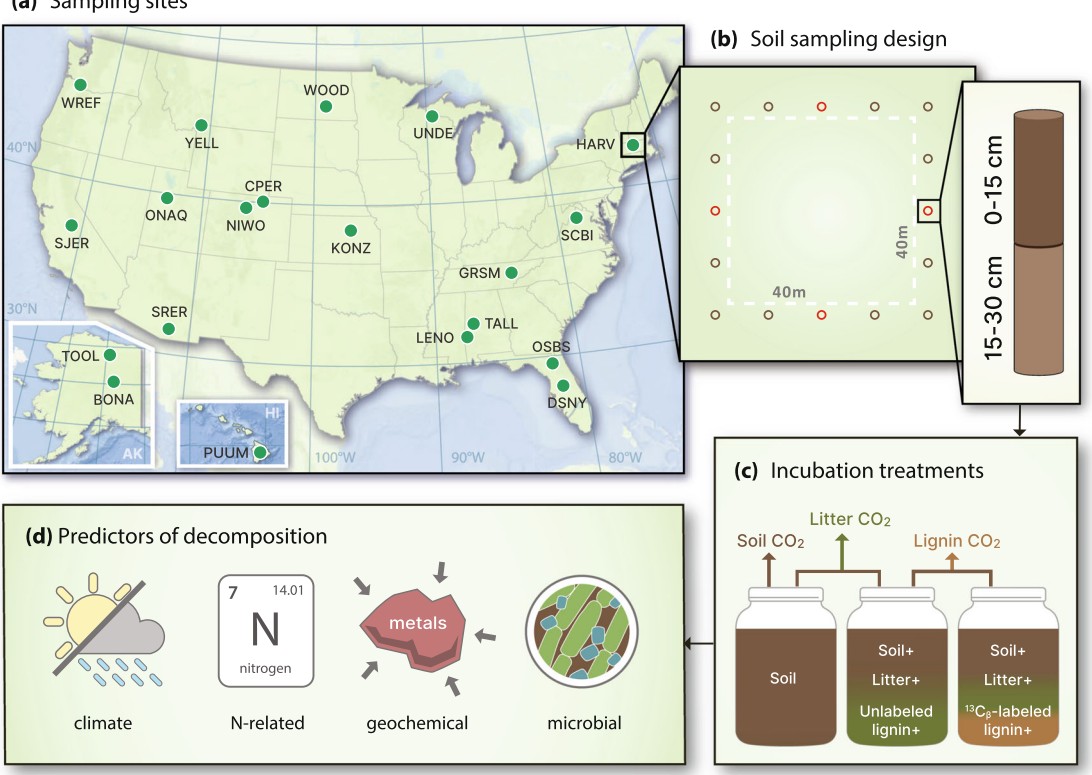

**Fig. 1 | Sampling sites and experimental design of this study. a** The 20 NEON sampling sites; **b** soil sampling points around a 40 × 40-m plot at each site; **c** experimental design to partition sources of C decomposition; and **d** biogeochemical predictors, which included climate, N-related, geochemical, and microbial variables (Supplementary Table 2). For lab incubation, four mineral soils at 0–15 and 15–30 cm (e.g., points in red, **b**) were incubated with three substrate treatments (soil alone, soil + C4 litter + unlabeled lignin, and soil + C4 litter + $^{13}C_\beta$-labeled lignin) to partition C decomposition among lignin (orange), litter (green), and SOC (brown) using C stable isotope measurements of $CO_2$. For field incubation, mesh bags with litter + unlabeled lignin or litter + $^{13}C_\beta$-labeled lignin were buried and retrieved after ~1 y to quantify cumulative lignin decomposition.

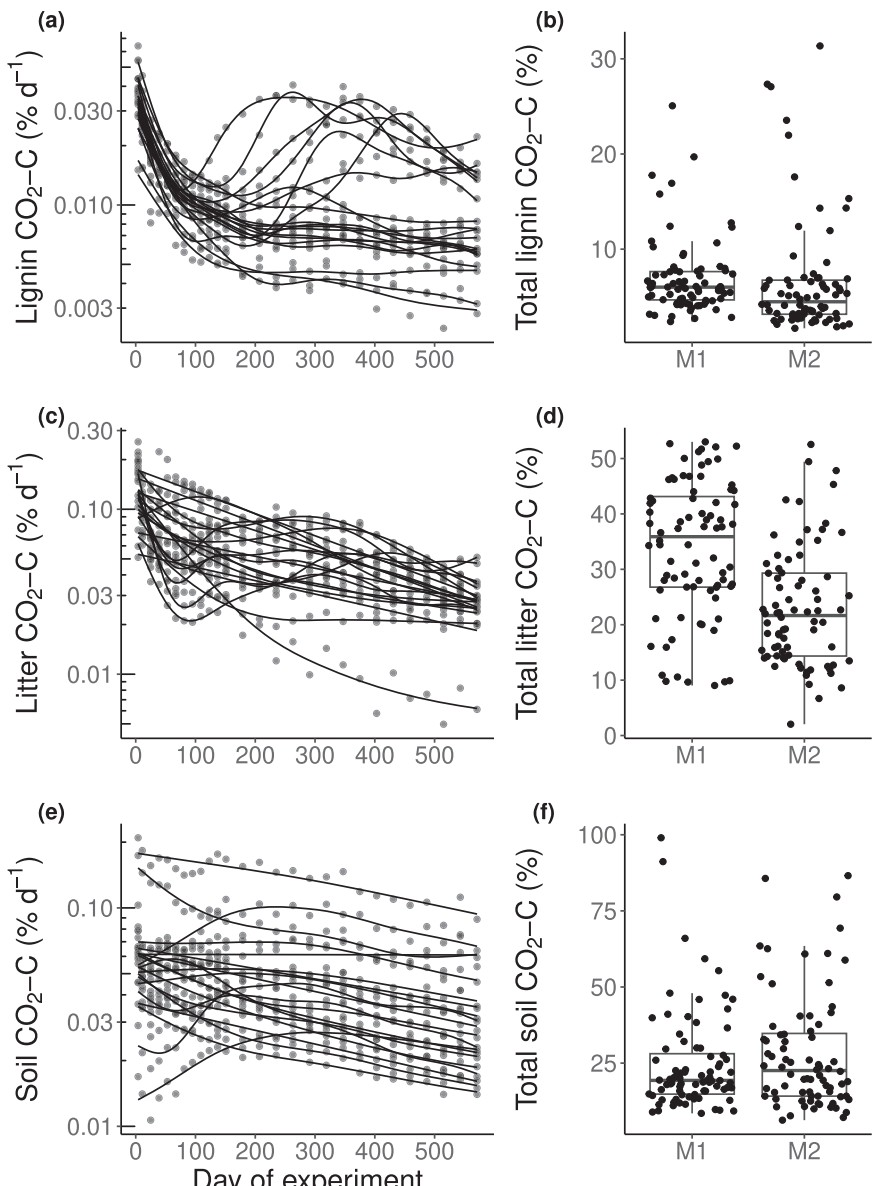

**Fig. 2 | Carbon (C) decomposition ($CO_2$-C) from lignin, litter, and soil in lab-incubated samples from 20 NEON sites, expressed as a percentage of the initial C mass in each pool.** Note the base-10 logarithmic $y$ axis scale for C decomposition rate (**a**, **c**, **e**). M1 and M2 denote mineral soil samples from 0–15 and 15–30 cm, respectively. Total C decomposition ($CO_2$-C) represents cumulative C decomposition over 18 months. Lines are fit by generalized additive mixed models (GAMMs). Each point in left panels (**a**, **c**, **e**) represents mean decomposition rate from four sampling points across two soil depths at each site ($n = 8$ biologically independent samples for each site, except for $n = 4$ for KONZ). The lines in the boxes are median values, and the edges of the boxes represent the 25th and 75th percentiles ($n = 156$ biologically independent samples). The upper and lower whiskers extend from the edge to the largest and smallest value no further than 1.5 times the interquartile range, respectively. Each point in the right panels (**b**, **d**, **f**) represents cumulative decomposition from each sampling point at each site. Source data are provided as a Source Data file.

and SOC decomposition over time (Fig. 1c), and to assess their relationships with climatic, N-related, geochemical and microbial factors across soils. Lignin decomposition is also measured in a separate N addition lab experiment and in field-incubated samples. We hypothesize that (1) lignin decomposition predictably varies with soil geochemical characteristics (reactive minerals and metals) and fungal communities at the continental scale, in addition to climatic and N-related variables, and that (2) the predictors for lignin decomposition are similar with litter decomposition, while these predictors have different relationships with SOC decomposition due to specific interactions of lignin with metals and microbial communities. Our results support these hypotheses that partially reconcile aspects of classic and modern views of decomposition, such that lignin decomposition is a

bottleneck for litter decomposition but not for SOC decomposition, thus explaining the variable contributions of lignin to SOC among soils as a function of their biogeochemical characteristics.

## Results

### Decomposition rates of lignin, litter, and SOC

The temporal dynamics of lab lignin C decomposition were generally more variable than litter C or SOC decomposition, although instantaneous lignin decomposition rate was about 4-fold lower than litter and soil decomposition, on average, when normalized by C mass in these pools (Fig. 2). Lab lignin decomposition rate generally decreased over time; however, in some soils, it transiently increased over timescales of months or was still increasing at the end of incubation (after

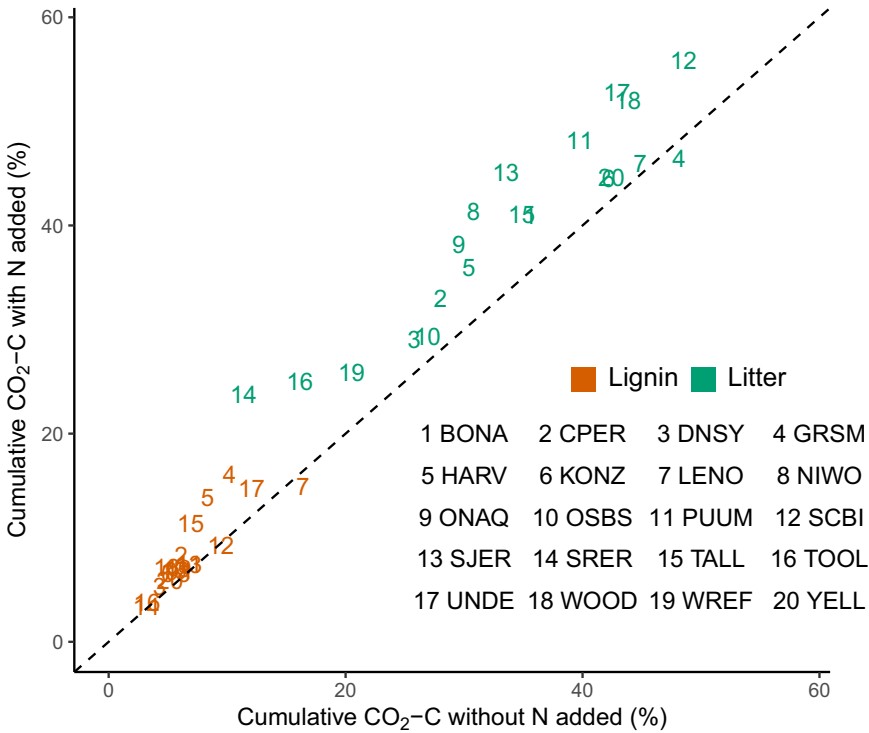

**Fig. 3 | Effects of nitrogen (N) addition on cumulative decomposition ($CO_2$-C) of lignin and litter incubated in the lab over 18 months, averaged from four sampling points at 0–15 cm depth from 20 NEON sites.** The decomposition is expressed as a percentage of the initial C mass in each pool. Numbers correspond to means from each NEON site according to the legend, denoted by four-letter site IDs (Fig. 1 and Supplementary Table 1). Numbers in orange and green denote the decomposition of lignin and litter, respectively. Each point is the mean of $n = 4$ biologically independent samples. Source data are provided as a Source Data file.

18 months) (Fig. 2a and Supplementary Fig. 2). Litter decomposition rate also increased transiently over time in some sites (Fig. 2c), and it was significantly related to instantaneous lignin decomposition rate as indicated by Pearson correlation ($r = 0.65$; $P < 0.01$). The temporal pattern of lab lignin C decomposition differed from SOC decomposition, which generally showed a declining trend at the end of the incubation (Fig. 2e); instantaneous decomposition of lignin C and SOC were not statistically related ($r = -0.04$; $P > 0.05$). At the end of the lab incubation (571 d), cumulative C decomposition relative to initial C was 1.7–31.4% for lignin, 2.0–53.0% for litter, and 6.3–99.0% for SOC (Fig. 2b, d, f). The soils with coolest climate (TOOL, a Gelisol) showed the lowest site-averaged decomposition of lab lignin (3.1%), litter (15.4%) and SOC (13.3%) relative to other soils. The soils with warm and dry climate (ONAQ and SRER, Aridisols) had relatively lower site-averaged lab lignin (4.1%) and litter C (19.6%) decomposition but the highest site-averaged SOC decomposition (49.4%). The cumulative decomposition of lignin and SOC was similar between 0 and 15 cm and 15–30 cm soil samples, while cumulative litter decomposition was significantly lower in the deeper soil (34% for 0–15 cm vs. 23% for 15–30 cm; $P < 0.01$).

We conducted a separate N addition experiment to test effects of N availability on lignin and litter decomposition in the 0–15 cm soils. Lab lignin and litter decomposition rates were slightly increased by N addition in most sites throughout the lab incubation (Supplementary Fig. 3), and cumulative decomposition was significantly greater after the 571-d incubation ($P < 0.01$ for both; Fig. 3). On average, N addition increased cumulative lab lignin decomposition by 1.6% and litter decomposition by 6.2%, and neither lignin nor litter decomposition was significantly depressed by N addition at any site after 18 months (Fig. 3).

Our field sites had large climate differences whereas the lab samples were incubated at the same temperature and comparable moisture, so we used an additional field lignin decomposition

experiment with 0–15 cm soils to test whether similar biogeochemical predictors were important in the field and in the lab. The field experiment was conducted in mesh bags that allowed additional microbes to colonize the soil/litter/lignin mixtures over time. Cumulative field lignin C decomposition after ~1 y showed higher variation among samples and overall higher rates than observed in the lab (Supplementary Fig. 4), corresponding with overall higher fungal quantity in the field than in the lab (Supplementary Fig. 1). Total field lignin C loss relative to initial lignin C concentrations averaged 9–63% across the 20 sites, while the site-averaged lab lignin C loss was 3–16% (Supplementary Fig. 4). There was no significant correlation ($P > 0.05$) between field and lab lignin C loss.

**Variation in biogeochemical predictors among soils**
Along with climatic factors (mean annual temperature, MAT, and mean annual precipitation, MAP), we selected 25 biogeochemical predictors and separated them into three categories, including those related to N availability (11), geochemistry (8), and microbes (6) (Supplementary Table 2 and Fig. 1). The variation of these biogeochemical predictors was very high among sites and was generally similar between the lab and field incubation datasets (Supplementary Fig. 1). For the N predictors, total soil N was 0.1–32 mg $g^{-1}$ and C/N was 8–58. Both $NH_4^+$-N (0–104 µg N $g^{-1}$) and $NO_3^-$-N (0–937 µg N $g^{-1}$) tended to increase with incubation time in the lab (Supplementary Fig. 1a). For the geochemical predictors, soil pH was 4.0–9.2, and soil particle size (silt+clay, 5–91%) and metals had at least one order of magnitude difference among sites (0–30 mg $g^{-1}$ $Al_{ox}$; 0–20 mg $g^{-1}$ $Fe_{HCl}$, 0–78 mg $g^{-1}$ $Fe_{ox}$; 0–54 mg $g^{-1}$ $Fe_{cd-ox}$, 0.0–3.3 mg $g^{-1}$ $Mn_{cd}$, 0–55 mg $g^{-1}$ $Ca_{cd}$; Supplementary Fig. 1b). Microbial community variables exhibited as much as four orders of magnitude difference among sites, with fungal quantity of $1.1 \times 10^5$–$2.6 \times 10^9$ gene copies $g^{-1}$, bacterial quantity of $5.2 \times 10^9$–$9.8 \times 10^{11}$ gene copies $g^{-1}$, fungal-to-bacterial ratio of $5.8 \times 10^{-6}$–$6.2 \times 10^{-2}$, and a fungal diversity index of −39–73 (i.e., the

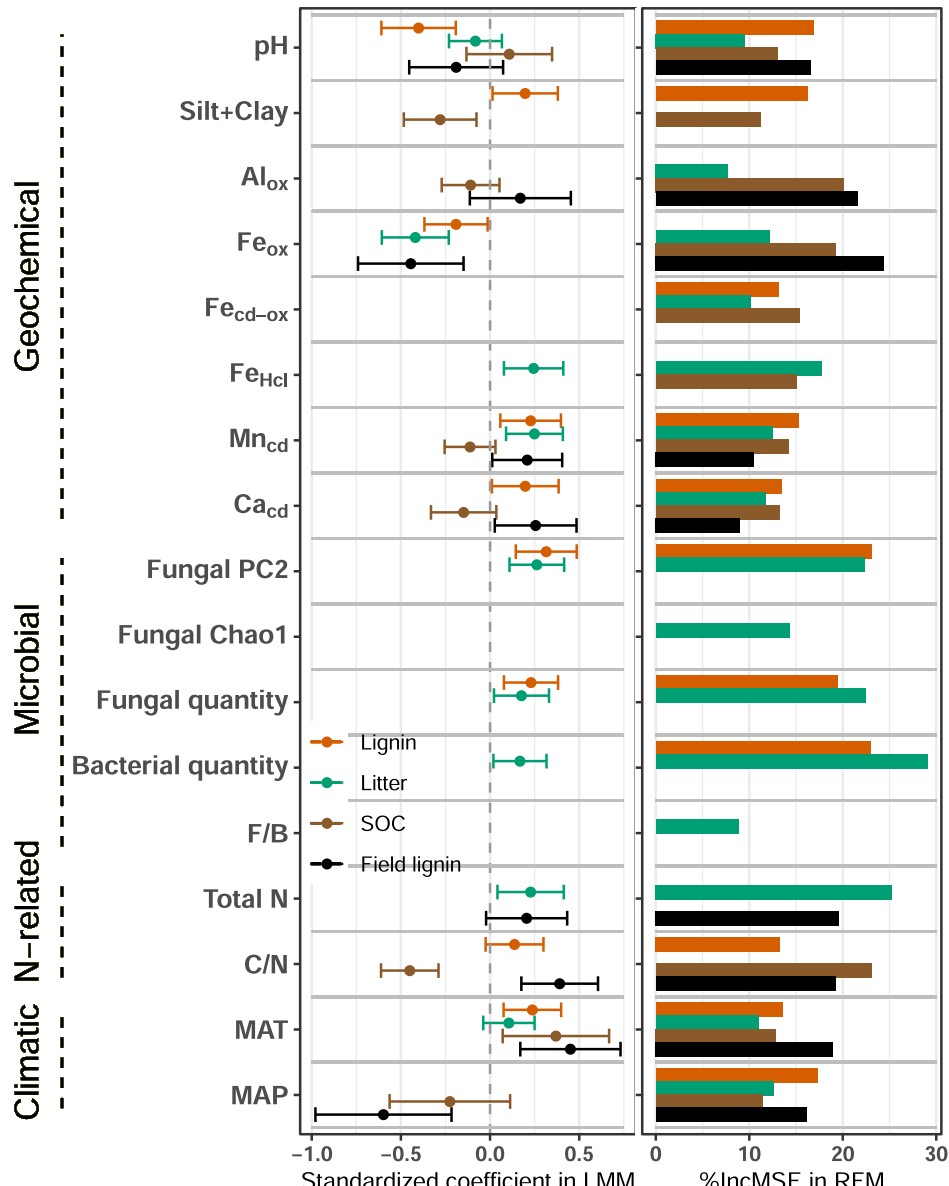

**Fig. 4 | Effects of predictors on cumulative lignin, litter, and soil organic carbon (SOC) decomposition estimated from linear-mixed models (LMM, left) and random forest models (RFM, right).** Predictors correspond to those in Supplementary Table 2. $R^2_{fixed}$ and $R^2_{model}$ represent variance explained by fixed effects and fixed + random effects in the LMM, respectively; $R^2_{RF}$ represents variance explained by the RFM. For lignin decomposition (orange, $n = 155$ biologically independent samples), $R^2_{fixed} = 0.43$, $R^2_{model} = 0.45$, $R^2_{RF} = 0.51$; for litter decomposition (green, $n = 155$ biologically independent samples), $R^2_{fixed} = 0.49$, $R^2_{model} = 0.52$, $R^2_{RF} = 0.60$; for soil C decomposition (brown, $n = 156$ biologically independent samples), $R^2_{fixed} = 0.43$, $R^2_{model} = 0.71$, $R^2_{RF} = 0.55$; for field lignin decomposition (black, $n = 151$ biologically independent samples), $R^2_{fixed} = 0.31$, $R^2_{model} = 0.53$, $R^2_{RF} = 0.39$. Estimated standardized regression coefficients from the LMMs are plotted with error bars representing 95% confidence intervals (±2 standard error). The %IncMSE in the RFMs show the increase of the mean squared error when a given predictor is randomly permuted; the larger the value, the more important the predictor. Source data are provided as a Source Data file.

residual of the Chao1 index for fungal communities, Supplementary Fig. 1c). Microbial quantity changed with time in the lab-incubated samples: fungal quantity increased after 9 months vs. the initial soil samples, and fungal quantity and bacterial quantity also increased after 14 months vs. 9 months (Supplementary Fig. 5a).

**Importance of biogeochemical predictors for C decomposition**
We used three statistical approaches (linear-mixed models, LMMs, generalized additive mixed models, GAMMs, and random forest models, RFMs) to identify the most important soil geochemical, microbial, N, and climatic predictors of decomposition (Fig. 4). The RFM partial dependence plots showed that many relationships between predictors and response variables were approximately linear

until predictors increased above the 90th or larger percentiles, where response variables became approximately constant (Supplementary Figs. 6–9). The GAMMs similarly demonstrated that nearly all predictors had linear relationships with response variables (Supplementary Table 3). Most of our optimal statistical models included multiple predictor variables from most categories (N-related, geochemical, and microbial variables; Fig. 4). The optimal RFMs included similar predictors as the LMMs, with a few exceptions (see Supplementary Notes). Thus, for clarity of explanation we hereafter focused mainly on results from the LMMs. In addition, to test whether biogeochemical predictors changed with time throughout lab incubation, we compared models of cumulative decomposition after 6, 12, and 18 months (Supplementary Figs. 10 and 11). Predictors were generally similar over

time, and thus we focused our subsequent analysis on the 18-month (571 d) dataset (the other results are presented in the Supplemental Information).

The LMM of lab lignin decomposition showed that soil pH and fungal composition were the strongest predictors, and that MAT, fungal quantity, $Mn_{cd}$, $Ca_{cd}$, silt+clay, $Fe_{ox}$, and soil C/N could also improve the final model (Fig. 4). Soil pH and $Fe_{ox}$ had negative relationships with lab lignin decomposition while all other predictors had positive relationships. These predictors explained 43% of the observed variance in lab lignin decomposition; the overall model (including random effects for site) explained 45%. Lignin decomposition in the field and lab shared many of the same predictors in the LMMs, including MAT, $Fe_{ox}$, $Ca_{cd}$, and $Mn_{cd}$ (Fig. 4). The $Fe_{ox}$ showed a negative relationship while the other three predictors showed positive relationships with field lignin decomposition. These four predictors, along with MAP, soil C/N, soil N, pH, and $Al_{ox}$, explained 31% of the variation in field lignin decomposition; the overall model with random effects explained 53% of the variation.

Predictors of lab litter decomposition were generally similar to lab lignin decomposition. The LMM showed that $Fe_{ox}$ was the strongest predictor of litter decomposition, followed by fungal composition, $Mn_{cd}$, $Fe_{HCl}$, soil N, fungal quantity, bacterial quantity, MAT, and pH (Fig. 4). $Fe_{ox}$ and pH were negative while all other predictors were positively related to litter decomposition. These predictors explained 49% of the variation in litter decomposition, and the overall model with random effects explained 52%.

Contrary to lignin and litter decomposition, microbial variables were not important predictors of SOC decomposition in the statistical models (Fig. 4). The LMM showed that soil C/N, MAT, and silt+clay were the strongest predictors for SOC decomposition, and that MAP, $Ca_{cd}$, $Mn_{cd}$, $Al_{ox}$, and pH were also important (Fig. 4). MAT and pH were positively while all other variables were negatively related to SOC decomposition. The predictors collectively explained 43% of the variation in SOC decomposition and the overall LMM (including random effects) explained 71%.

## Discussion

Overall, our continental-scale data showed the particular importance of geochemical and microbial predictors for lignin and litter decomposition, and their differing relationships with SOC decomposition (Fig. 4), consistent with our first and second hypothesis, respectively. Our results collectively supported different aspects of classic and modern views of decomposition. The strong correlation between lignin and litter decomposition (Fig. 2) and the similar biogeochemical predictors of these processes support the classic view that lignin decomposition is tightly coupled with overall litter decomposition[1,27]. However, we found that decomposition of SOC was unrelated to decomposition of lignin and litter, and that these processes often had contrasting relationships with biogeochemical predictors. Several soil geochemical factors had negative ($Fe_{ox}$ and $Al_{ox}$) and positive ($Fe_{HCl}$, $Mn_{cd}$, and $Ca_{cd}$) relationships with lignin and/or litter decomposition, while there were almost entirely negative relationships between extractable soil metals and SOC decomposition (Fig. 4; Supplementary Fig. 10). Intriguingly, microbial variables including fungal composition and fungal and bacterial quantity were needed to explain variation in the decomposition of lignin and litter, but not SOC (Fig. 4). These findings are inconsistent with the classic idea that the slow decomposition of lignin residues limits decomposition of total SOC[3]. Rather, the disparate rates and predictors of lignin and SOC decomposition support the modern proposal that lignin depolymerization is not necessarily a primary bottleneck for SOC decomposition[5,7]. Furthermore, our dataset provides an explanation for the decoupling of lignin and SOC decomposition by highlighting their differing relationships with geochemical and microbial variables.

Our data demonstrated that lignin decomposition was not universally slow or fast when compared with decomposition of litter and SOC, but rather, it varied predictably among sites along with biogeochemical variables (Fig. 4). Long-term climatic predictors (MAT and MAP) could explain variation in both field and lab decomposition, but the actual vs. legacy climate, as reflected by the field vs. lab experiments, respectively, had different relationships with lignin decomposition (Fig. 4; Supplementary Figs. 6 and 7). A relatively high amount of N addition stimulated decomposition of lignin and litter to a minor degree in most sites after 18 months (Fig. 3), but its overall impact was small when considering the wide range of decomposition across samples (Fig. 2). Inorganic N appeared to be less important than geochemical and microbial properties (Fig. 4).

We found that geochemical variables often had different relationships with lignin and litter decomposition than with SOC decomposition (Fig. 4). For lignin and/or litter decomposition, some geochemical variables had negative relationships (e.g., $Fe_{ox}$ in the LMM and RFM and $Al_{ox}$ in the RFM) and others had positive relationships (e.g., $Fe_{HCl}$, $Mn_{cd}$ and $Ca_{cd}$ in the LMM and RFM), while for SOC decomposition, they mainly had negative relationships (e.g., $Al_{ox}$, $Mn_{cd}$, and $Ca_{cd}$ in the LMM and RFM, and $Fe_{HCl}$ and $Fe_{cd-ox}$ in the RFM). The metals extracted from these NEON soils likely represent ions (e.g. $Ca_{cd}$), metals dissolved from mineral phases of varying crystallinity (e.g., $Fe_{HCl}$, $Fe_{ox}$, $Fe_{cd-ox}$), or a mixture of ions and mineral phases (e.g., $Mn_{cd}$ and $Al_{ox}$)[32]. Synthesis studies and lab experiments demonstrate that soil mineral and metal cations as well as fine particles (silt+clay) are important predictors of SOC concentration due to protection by sorption, precipitation, and aggregation[16,20,33]. In our study, protective effects of soil metals and minerals were mainly applicable for SOC decomposition, and conversely, some of these same variables were actually associated with greater decomposition of lignin and litter.

We found positive associations of some soil metals (e.g., $Mn_{cd}$ and $Ca_{cd}$ in the LMM, and $Fe_{HCl}$ in the RFM) with lignin and litter decomposition (Fig. 4 and Supplementary Fig. 10), consistent with catalytic or biological roles of soil metals for organic matter decomposition demonstrated in other studies[19,21,34]. The finding that $Fe_{HCl}$ was positively related and $Fe_{ox}$ was negatively related with lignin and litter decomposition was consistent with multiple functional roles of Fe, which might stimulate decomposition or provide protection depending on C molecular composition and/or redox environment[13,34]. Moreover, lignin and litter decomposition increased in samples with greater Mn and Ca (Fig. 4), consistent with the importance of Mn-promoted degradation of organic C[18]. Mn can promote lignin decomposition via enzymes and redox cycling[35], which may have increased overall litter decomposition[19]. The strong positive relationships between Ca and decomposition of lignin and litter agreed with previous studies showing that Ca was positively related to the extent of litter mass loss, and in particular, lignin degradation[21,36] as Ca is an essential component of the fungal cell wall and can increase the growth of white rot fungi[37]. We also found an overall positive relationship of silt+clay with lignin and litter decomposition, which might reflect multiple biological and physical factors that co-vary with particle size, as well as the potential for minerals to catalyze OM decomposition[38]. Overall, the role of certain metals and fine particles in stimulating lignin and litter decomposition while suppressing SOC decomposition provides an explanation for the fact that these processes may be coupled or decoupled to varying degrees[39], depending on soil characteristics.

Consistent with our hypotheses, composition, and quantity of overall fungal communities explained variation in lignin and litter decomposition (Fig. 4). Intriguingly, however, only three of the fungal genera significantly correlated with lab lignin decomposition have been reported to degrade lignocellulose (*Trichocladium*, soft-rot; *Mycena* and *Hypochnicium*, white rot)[40–42] (Supplementary Table 4). This indicates that the most commonly studied lignin-degrading fungi

(i.e., the known "rot" fungi) were not necessarily the most important lignin-degrading organisms in our continental-scale dataset[25]. Consistent with a previous study across North America[43], we found that fungal communities were highly heterogenous across sites and even within plots; e.g., only 65 of 342 fungal species occurred in >10 samples. This finding further suggests that specific fungal taxa possibly responsible for lignin decomposition varied with locations and even depths at the same plot. Bacterial quantity was also related to both lignin and litter decomposition (Fig. 4 and Supplementary Fig. 10). Although bacteria may degrade lignin directly[23], they might simply be responding to increased C availability as a consequence of fungal lignin decomposition, or may synergistically interact with fungi to promote lignin decomposition[44]. Our findings build on previous laboratory experiments that demonstrated an impact of microbial composition on decomposition of inoculated litter[45,46], by showing that fungal community composition and abundance explained variation in lignin and litter decomposition rates even among diverse soils. This challenges the hypothesis of microbial functional redundancy[22,47]. Furthermore, the differing relationships between fungal composition and decomposition of lignin and SOC provide another explanation for the observed decoupling of these processes.

Contrary to lignin and litter decomposition, microbial variables were less important predictors of SOC decomposition (Fig. 4), despite high variation in fungal composition and richness and bacterial and fungal quantities across soils (Supplementary Figs. 1 and 5). Different axes of fungal community composition correlated with decomposition of lignin and litter (PC1) vs. SOC (PC2; Supplementary Fig. 10). Significant relationships among SOC decomposition rate and microbial community composition, biomass, and richness have sometimes been reported[48], but other studies found weak relationships[49]. Consistent with the latter findings, we found that microbial predictors including fungal community composition, fungal quantity, and bacterial quantity were not related to SOC decomposition after accounting for other variables (Fig. 4). One possible explanation for the null relationship between microbial predictors and SOC decomposition is that SOC turnover is dominantly determined by decomposer access to SOC[26]. A large proportion of the SOC in our incubated soils was probably stored in small pores inaccessible to microbes[50] that did not protect the added lignin and litter, which were gently mixed into the soil. This might explain why decomposition of the added lignin and litter was measurably related to fungal community composition and quantity, whereas decomposition of SOC was not.

After accounting for other biogeochemical predictors, MAT and MAP of the study sites was still related to organic matter decomposition (Fig. 4) even under the common conditions of temperature and moisture imposed in the lab incubation. This is consistent with previous findings that climate history influenced litter and SOC decomposition, possibly by shaping the composition and functional responses of decomposer communities and/or via correlation with soil minerals through secondary mineral formation[10,51,52]. Microbial communities from different soils can remain distinct over months to years even when exposed to a common temperature and moisture regime, and in spite of changes in community composition over time[45,46] (e.g., Supplemental Fig. 5). Climate greatly impacts soil weathering[9,10], and although our statistical models included geochemical variables, it is also possible that the apparent relationships between decomposition and climate also reflected geochemical differences that were not accounted for by the extractable metals data (Fig. 4). It was not surprising that MAP had different relationships with field and lab lignin decomposition (Fig. 4, Supplementary Figs. 6 and 7), given that MAP reflected either the actual differences in climate during the field experiment or the legacies of prior differences in climate during the lab experiment. Nevertheless, all organic matter decomposition variables were positively related to MAT in the statistical models

(Fig. 4), suggesting that the legacy effect of soil MAT was stronger than the legacy effect of MAP on OM decomposition.

On balance, inorganic N addition led to only a small net stimulation of lab lignin and litter decomposition after 18 months (Fig. 3), and the effects of N were relatively small in comparison with variation across sites (Figs. 2 and 3). The positive response of lignin and litter decomposition to N addition might imply that microbial growth was N-limited in many sites. However, lab lignin and litter decomposition were not consistently related to inorganic N in the experiment without N addition, and they had differing relationships with total N and C:N (Fig. 4, Supplementary Figs. 6, 8 and 10). When considering continental-scale variation in biogeochemical properties, variation in N availability may be a less important driver of decomposition than sometimes assumed.

Comparison of our results with other recent observations from NEON soils indicates that the differing controls on decomposition of lignin and litter vs. SOC may contribute to variation in SOC concentration and organic matter composition among ecosystems. Many of the same variables that predicted C decomposition in the lab incubation also predicted differences in SOC concentration and the distribution of SOC between size fractions, defined as chemically dispersed particulate organic C (>53 µm, likely derived mostly from plants) and mineral-associated organic C (<53 µm, likely derived from a mixture of plants and microbes), which were described in previous studies of NEON soils[33,53]. Silt- and clay-sized minerals and reactive Fe phases in particular have long been thought to protect SOC from decomposition, even though the relationships among these variables can be relatively weak across large datasets[33,54]. Here, we found that the magnitude or even the sign of the pairwise correlation or model coefficient between decomposition and silt+clay or Fe in various extractions ($Fe_{HCl}$ and $Fe_{ox}$) often differed between lignin/litter and SOC (Fig. 4 and Supplementary Fig. 10).

These differences could influence SOM composition while explaining the context-dependency of relationships between Fe and SOC concentration in other datasets[33,54]. For example, negative relationships of $Fe_{ox}$ with lignin and litter decomposition (Fig. 4) could help explain the positive relationship between $Fe_{ox}$ and the increasing proportion of SOC in particulate vs. mineral-associated organic C, which we observed in our previous study with the same soils[53]. That is, $Fe_{ox}$ could increase particulate organic C by disproportionately decreasing rates of lignin decomposition relative to bulk SOC, consistent with the view that particulate organic C is mostly composed of decomposing plant detritus which may aggregate with certain metals[55]. Similarly, the positive relationship between silt+clay and lignin decomposition and its negative relationship with SOC decomposition is consistent with our previous finding that increased silt+clay was associated with lower particulate vs. mineral-associated organic C[53]. This might simply be due to increased capacity for mineral protection, but it might also be linked to increased catalysis of lignin decomposition by metals and/or minerals in these fine particle fractions[38]. Together, the contrasting relationships of silt+clay and $Fe_{ox}$ with decomposition of lignin and litter vs. SOC provide an explanation for why these variables may be poor predictors of SOC concentration over broad scales, even though they may be related to the physical forms of SOC (particulate vs. mineral-associated organic C).

In summary, using a quantitative isotopic method, we found that decomposition of lignin varied 18-fold among soils sampled from sites across North America and incubated in a common environment. Lignin decomposition was always slower than but was strongly related to bulk litter decomposition. Differences in lignin decomposition among sites were strongly related to biogeochemical predictors, in a manner that was similar to bulk litter decomposition but often differed from SOC decomposition. Different axes of fungal community composition were related to decomposition of lignin and litter compared to SOC, and metals often positively correlated with lignin decomposition even

though they had a neutral or negative correlation with SOC decomposition. Similarities in controls on lignin vs. bulk litter decomposition reinforce the traditional view that lignin is tightly coupled with overall litter decay over timescales of months to years, although this might differ in environments subject to photodegradation[56]. In contrast, the difference in controls on lignin and litter decomposition vs. SOC supports the modern notion that lignin depolymerization is not always a primary bottleneck for SOC decomposition. Based on the observed differences in drivers of lignin decomposition vs. SOC decomposition, we might expect lignin to be a more important component of SOC in soils with higher $Fe_{ox}$ and lower $Mn_{cd}$, $Ca_{cd}$, and silt+clay. Decomposition of all C forms increased with site MAT even though samples were incubated under a common temperature, possibly reflecting microbial or geochemical legacies related to climate. While substantial research has focused on N dynamics as controls on litter decomposition, our data showed that the influence of N availability on decomposition of lignin and litter was often smaller than other geochemical and microbial factors. Together, our data demonstrate the critical need for mechanistic models to account for contrasting geochemical and microbial controls on decomposition of lignin and litter vs. SOC, in addition to the traditional variables of climate, residue quality, and nutrient availability.

## Methods

### Experimental design

We used 20 sites from National Ecological Observatory Network (NEON) to examine decomposition of lignin, bulk litter, and SOC and to test biogeochemical predictors of decomposition of these substrates, including geochemical, microbial, N-related, and climatic variables. Soils amended with C stable isotope ($\beta$-$^{13}$C)-labeled and unlabeled lignins[57] and a single natural litter source were incubated in the lab to quantify lignin, litter, and SOC decomposition over 18 months (Fig. 1c). An additional incubation experiment was also conducted to test the effects of N addition on lignin and litter decomposition. The results of lignin decomposition and its predictors from the lab incubation were further compared with those from a field incubation. The lab incubation enabled us to compare C decomposition among samples while standardizing temperature and moisture, whereas the field incubation allowed us to assess effects of actual site temperature and moisture on lignin decomposition, while also allowing for sample colonization by additional microbes.

### Site selection and soil sampling

NEON is a U.S.-based, continental-scale ecological monitoring network that provides open data, samples, and research infrastructure to reveal how ecosystems are responding to environmental change[58]. NEON sites are stratified among domains defined by climate characteristics[58], not by soil type, and while they naturally contain a wide diversity of soil types, soils at each site are not necessarily representative of the corresponding ecoclimatic domain. For this project, we selected 20 NEON terrestrial sites, denoted by their acronyms as follows: BONA, CPER, DSNY, GRSM, HARV, KONZ, LENO, NIWO, ONAQ, OSBS, PUUM, SJER, SRER, SCBI, TALL, TOOL, UNDE, WREF, WOOD, YELL (Fig. 1a). These sites span wide edaphic, climatic and ecosystem gradients (Supplementary Fig. 1), and they were chosen to span broad differences in biogeochemical characteristics, within constraints of feasibility and permitting. They encompass 9 out of the 12 soil orders in the United States Department of Agriculture (USDA) soil classification system (no Histosols, Oxisols, or Vertisols; Supplementary Table 1). The nine soil orders include Alfisols (CPER and SCBI), Andisols (WREF), Aridisols (ONAQ and SRER), Entisols (DSNY, OSBS and PUUM), Gelisols (TOOL), Inceptisols (BONA, GRSM, HARV, LENO and NIWO), Mollisols (KONZ, SJER, WOOD and YELL), Spodosols (UNDE), and Ultisols (TALL). The sites had mean annual temperature (MAT) of −9–22 °C and received 262–2657 mm of mean annual precipitation (MAP). The sites included

diverse ecosystem types, such as tundra, forest, wetland, grassland, shrubland, and desert.

Soils at each site were sampled by NEON staff during the growing season of 2019 (April–August; later sampling occurred at Alaska sites where soils did not thaw until July or August). Mineral soil samples were collected at two depths (0–15 cm and 15–30 cm), after removing any surface litter or organic horizon (Supplementary Table 1), using a 2- to 5-cm diameter corer, according to the standard NEON sampling procedure for that particular site. At each site, samples were collected around the perimeter of one $40 \times 40$-m "distributed base plot" selected to represent the dominant upland vegetation and soil type of that site whenever possible. Soil at the KONZ site was collected only at 0–15 cm due to the shallow soil depth. Each plot had 16 replicates ($n = 16$), denoted sampling points 1–16 hereafter. Point 1 was located 4 m west and 4 m south from the SW corner of each plot, and the other points were located in counterclockwise sequence at 12-m intervals around the perimeter of the plot, each located 4 m outside of the plot boundary (Fig. 1b). Soil cores from each point were collected and shipped overnight on ice (~4 °C) to Iowa State University (ISU) for use in laboratory and field incubations. Soil from each sample was gently homogenized inside a plastic bag after any coarse roots, macrofauna, or rocks were manually removed. We did not sieve samples except for ONAQ and SRER, where rocks were abundant and were removed by passing soil through a 2-mm sieve.

### Lab incubation experiments

Soils from four sampling points at two depths per site were used for lab incubation and biogeochemical analyses, totaling 156 samples. The four sampling points were mainly selected at the odd number in the middle of each side of the $40 \times 40$-m plot (red circles in Fig. 1b), although sampling points from some sites were selected at the numbers next to the middle numbers if soils were not available for both layers. Subsamples of soils used for the lab incubation experiment were brought to field moisture capacity, which was determined for each soil by saturating an additional 20–30-g subsample placed on filter paper in a funnel, and measuring gravimetric water content following 48 h of drainage. Subsamples (1 g dry mass equivalent) from each sampling point and depth were incubated under each of three separate substrate treatments to partition C decomposition among three sources, using measurements of $\delta^{13}C$ values of $CO_2$. We quantified decomposition of C from extant soil organic matter, C from added litter (senesced leaves of *Andropogon gerardi*, a $C_4$ grass), and a specific C atom (the $C_\beta$ position of the propyl sidechain) in lignin that was precipitated on the added litter. The lignin was prepared as described in the Supplementary Methods.

Substrate treatments were: (1) soils alone (control); (2) soils amended with *A. gerardi* litter precipitated with trace natural abundance $^{13}C$ lignin (soil + litter + unlabeled lignin); (3) soils amended with *A. gerardi* litter precipitated with trace lignin labeled with 99 atom percent $^{13}C$ at the $C_\beta$ position of each lignin $C_9$ substructure (soil + litter + $^{13}C_\beta$-labeled lignin). We added uniform litter and synthetic lignin to each of the mineral soils to focus on soil biogeochemical gradients rather than substrate quality. Soils were gently mixed with the litter + lignin mixture in a 250:25:1 ratio of soil:litter:lignin (1 g dry soil mass equivalent was mixed with 100 mg litter and 4 mg lignin). To prepare the litter + lignin mixture, the unlabeled or labeled lignins were precipitated in a 1:25 mass ratio on dried and finely ground leaf litter of *A. gerardi* (41.9% C, 0.41% N, and $\delta^{13}C = -12.6‰$; see Supplementary Methods for more details). The 20 NEON sites comprise ecosystems ranging from $C_3$-dominated forest and grassland sites to mixed $C_3$–$C_4$ grasslands and/or plants with Crassulacean acid metabolism, such that the $\delta^{13}C$ value of the added $C_4$ litter was always more positive than $\delta^{13}C$ value of $CO_2$ derived from soil organic matter at a given site.

Soil samples were incubated under oxic conditions in the dark at 23 °C for 571 d. Soil was kept in an open 50 mL centrifuge tube inside a glass jar (946 mL) sealed with a gas-tight aluminum lid with butyl septa for headspace gas purging and sampling. The jars were flushed with $CO_2$-free air following periodic headspace sampling as described below, and $CO_2$ concentrations remained below 5000 ppm during the incubation. Assuming a 1:1 ratio between $CO_2$ production and oxygen ($O_2$) consumption, $O_2$ decreased by <2.4% of the initial value (20.9%) during each sampling period. Because the volume of incubated soil was ~1000-fold smaller than the jar headspace, $CO_2$ produced by soil microbes would diffuse out of the soil and accumulate in the headspace with negligible storage in soil pores. Soil moisture was monitored by recording the mass of each sample, and water was added as necessary to match the original mass of each sample under field moisture capacity every month before 179 d and every other month thereafter (due to the less frequent gas sampling) to replenish vapor lost during headspace flushing. To monitor instantaneous decomposition over time and to avoid accumulation of $CO_2 > 5000$ ppm in the jar, headspace gas was initially measured at 4 d and 11 d, every other week for another 140 d, and then every month after 179 d (for a total duration of 571 d). The $CO_2$ concentrations and their $\delta^{13}C$ values were measured by a tunable diode laser absorption spectrometer (TGA200A, Campbell Scientific, Logan, UT) immediately prior to flushing the headspace[57]. Because jars remained sealed between headspace sampling events, we were able to quantify the entire cumulative production of $CO_2$ and its $\delta^{13}C$ value from each replicate over the course of the experiment. The $CO_2$ production from soil was measured on samples with no addition of litter and lignin, and $CO_2$ from litter and $^{13}C_\beta$-labeled lignin was calculated by two-source mixing models that used measurements from the litter + unlabeled lignin and litter + $^{13}C_\beta$-labeled lignin treatments, respectively[13,57] (see Supplementary Methods for more details). The C decomposition from soil, litter and lignin were expressed as percentages of their initial C masses (41.9 mg for litter and 264 µg for the $^{13}C_\beta$ atom of the labeled lignin, and a variable amount for SOC; Supplementary Table 1).

We also conducted an N addition experiment to test the effects of N availability on lignin and litter decomposition, using additional subsamples of the 0–15 cm soils collected from the four sampling points described above. For this experiment, the subsamples amended with litter + unlabeled lignin or litter + $^{13}C_\beta$-labeled lignin were also amended with $NH_4NO_3$ at 50 mg N g$^{-1}$. The amount of added N was relatively high but comparable to inorganic N concentrations often observed in agricultural fields after fertilization[59]. Briefly, 51 mL of 0.0386 mol L$^{-1}$ $NH_4NO_3$ was added to soil samples, and then more water was added as necessary to achieve field moisture capacity. Sample incubation and gas measurements were the same as described above.

## Field incubation experiments

The 0–15 cm soils from all 16 sampling points at each site were used for field incubation (Fig. 1b). Soil subsamples (4.5 g dry mass equivalent) were gently mixed with litter + unlabeled lignin or litter + $^{13}C_\beta$-labeled lignin according to the mass ratios and substrate treatments described above. The soil + litter mixtures were then transferred to mesh bags (8 cm × 8 cm in size; 55 µm nylon screen), which allowed entry of fungal hyphae, bacteria, and soil microfauna while minimizing particle loss[2]. The mesh bags were sealed with hot glue and shipped back to the sites of origin and buried at a depth of 0–15 cm at the same locations where soils were initially sampled, and geo-referenced to facilitate retrieval. The mesh bags with litter + unlabeled lignin were buried at even-numbered sampling points for each site, and those with litter + $^{13}C_\beta$-labeled lignin were buried at the odd-numbered sampling points. After ~1 y of field incubation, the mesh bags were retrieved by NEON staff, flash-frozen on dry ice, and shipped on ice to ISU. Some bags

were damaged or could not be located in the field (31 out of 320 samples).

The soil and litter mixture was subsampled from each mesh bag, and then air-dried and finely ground for analysis of C concentrations and $\delta^{13}C$ at the UC Davis Stable Isotope Facility using an elemental analyzer (Elementar Analysensysteme GmbH, Hanau, Germany) and continuous flow isotope ratio mass spectrometer (Sercon Ltd., Cheshire, UK). Lignin C remaining after the 1-y field incubation was calculated by multiplying $f_{lignin}$ calculated based on a two-source mixing model (see details in Supplementary Methods) by the total C concentration in samples from the soil + litter + $^{13}C_\beta$-labeled lignin treatment, with corrections accounting for new C inputs as necessary based on measurements of the samples with unlabeled lignin (see details in Supplementary Methods).

## Soil inorganic N availability

We measured ammonium ($NH_4^+$) and nitrate ($NO_3^-$) in additional replicate soil + litter mixture samples (10:1 mass ratio of soil to litter) from all soils used in the lab incubation after 1, 9, and 18 months. Briefly, 10 g soil mixed with 1 g litter was placed in a 50 mL centrifuge tube, loosely covered, and then incubated at 23 °C in the dark after adjusting soil moisture to field capacity. Water was periodically added to soil samples to replace vapor loss, measured gravimetrically. Soil (~2 g) was subsampled from each centrifuge tube and extracted with 2 M potassium chloride at each timepoint. The soil solution was analyzed by microplate colorimetry for $NH_4^+$-N[60]. The $NO_3^-$-N was analyzed by microplate colorimetry[61] or for the 9-month samples, second-derivative spectroscopy[62]; these methods agreed almost perfectly on a subset of samples (slope = 0.95, $R^2 = 0.97$). Net N mineralization (sometimes known as potential N mineralization) was calculated as the difference in inorganic N between sets of sampling points (9-month vs. 1-month; 18-month vs. 9-month; 18-month vs. 1-month).

## Soil geochemical analysis

Most physical and geochemical measurements were made on soils from all of the sampling points used for the field and lab incubations, except for particle size and 0.5 M HCl extractions, which were done for the four sampling points per site used for laboratory incubation. Physical and geochemical measurements included soil pH, particle size fractions, 0.5 M HCl-extractable Fe(II) and Fe(III), ammonium oxalate-extracted metals (Al, Fe, Mn), and citrate dithionite-extracted metals (Al, Fe, Mn, and Ca). Some of these data were presented previously in a manuscript describing relationships between soil properties and particulate and mineral-associated organic matter fractions of these soils[53]. Field-moist soil subsamples were measured for pH in 1:1 slurries of soil and deionized water. Air-dried subsamples were used to measure particle size (sand, silt, and clay) by sieving and sedimentation following aggregate dispersion with sodium hexametaphosphate[53]. Field-moist subsamples were extracted with 0.5 M hydrochloric acid (HCl) to measure ionic Fe and highly reactive fractions of Fe(II) and Fe(III) minerals[63]. Concentrations of Fe(II) and Fe(III) were measured colorimetrically[63] and summed as $Fe_{HCl}$. Additional air-dried subsamples were extracted with acid ammonium oxalate in the dark at pH = 3 to measure organo-metal complexes and short-range-ordered (SRO) phases of Al, Fe, and Mn (denoted $Al_{ox}$, $Fe_{ox}$, and $Mn_{ox}$), and with sodium citrate dithionite to measure the crystalline and SRO phases of Fe ($Fe_{cd}$) as well co-occurring Al, Mn, and Ca ($Al_{cd}$, $Mn_{cd}$, and $Ca_{cd}$)[64]. Metals were analyzed via inductively coupled plasma optical emission spectrometry (PerkinElmer Optima 5300 DV, Waltham, MA). Extractions of Al and Mn by oxalate and citrate dithionite were very similar ($r = 0.88$ and $P < 0.001$ for Al; $r = 0.98$ and $P < 0.001$ for Mn), so we only report $Al_{ox}$ and $Mn_{cd}$. The difference between $Fe_{cd}$ and $Fe_{ox}$ represents crystalline phases ($Fe_{cd-ox}$). We interpret $Mn_{cd}$ as including exchangeable Mn, organo-metal complexes, and poorly crystalline phases. We

interpret $Ca_{cd}$ as a measure of exchangeable Ca and Ca in organo-Fe associations.

## Microbial analysis

DNA was extracted from soils for internal transcribed spacer (ITS) rRNA gene amplicon sequencing and quantitative PCR of 16 S and ITS rRNA regions. Each of the four soils per site used for lab incubation was subsampled for DNA extraction at the beginning of the incubation, and additional replicates were extracted after 9 and 14 months. The incubated replicates used for DNA extraction were prepared similarly to the replicates used for $CO_2$ analyses, and were amended with *A. gerardi* litter in a 1:10 mass ratio of litter to soil. The field-incubated soils corresponding to the same four sampling points for each site used in the lab incubation were also extracted for DNA, totaling 548 samples overall (156 soils × 3 timepoints for lab incubation and 80 soils for field incubation). Soils were stored at −80 °C before DNA extraction from 250 mg subsamples using the MagAttract PowerSoil DNA EP Kit (Qiagen, USA) on an Eppendorf epMotion 5075 liquid handling robot (Eppendorf North America, USA). Concentrations of DNA were measured using a Quant-iT™ dsDNA high-sensitivity Assay Kit (Invitrogen, USA) to standardize DNA masses for sequencing. Samples were diluted to 10 ng DNA $\mu L^{-1}$ prior to sequencing; samples with concentration <10 ng DNA $\mu L^{-1}$ were submitted directly. The ITS1 region of the ITS rRNA gene was amplified using the primer sets ITS1f (CTTGGTCATTTAGAGGAAGTAA) and ITS2 (GCTGCGTTCTTCATCGATGC), with PCR conditions as follows: 1 min at 94 °C, followed by 35 cycles of 30 s at 94 °C, 30 s at 52 °C and 30 s at 68 °C, and 10 min at 68 °C. Fungal ITS rRNA gene amplicon sequencing was performed on the Illumina Miseq platform at Argonne National Laboratory with library preparation using the Miseq Reagent Kit V2 (Illumina, USA), producing 2 × 250-bp reads.

Quantitative real-time PCR was performed on a CFX96™ real-time system coupled to a C1000™ thermal cycler (Bio-Rad, USA) to assess the quantity of 16 S and ITS rRNA genes. Each sample was prepared using 10 μL of SsoFast EvaGreen Supermix, 0.6 μL of each primer, 2 μL of diluted DNA sample, and nuclease-free water to a final volume of 20 μL. Bacterial 16 S rRNA genes were amplified using the primer sets 1055YF(ATGGYTGTCGTCAGCT) and 1392 R (ACGGGCGGTGTGTAC) and the following PCR conditions: 2 min at 50 °C and 10 min at 95 °C, followed by 40 cycles of 15 s at 95 °C and 1 min at 58 °C. Fungal ITS rRNA genes were amplified using the primer sets ITS1F_KYO1 (CTHGGTCATTTAGAGGAASTAA) and ITS2_KYO2 (TTYRCTRCGTTCTTCATC) and the following PCR conditions: 2 min at 50 °C and 2 min at 95 °C, followed by 40 cycles of 30 s at 95 °C, 30 s at 55 °C and 60 sec at 72 °C, and 10 min at 72 °C. Standard curves for 16 S and ITS rRNA genes were constructed using serial 10-fold dilutions from $10^{-1}$ to $10^{-8}$ of known concentrations of synthesized oligonucleotides (Integrated DNA Technologies, USA).

## Bioinformatics

We used the Divisive Amplicon Denoising Algorithm 2 (DADA2) pipeline[65] to process the ITS rRNA gene sequencing data in R statistical software version 3.6.1[66]. We excluded samples with ≤900 sequences, including 27, 78, and 1 sample collected after 0, 9, and 14 months of the lab incubation, respectively. All functions were run using default parameters suggested by the DADA2 pipeline tutorial. The end product included an amplicon sequence variant (ASV) table recording the number of times each exact ASV was observed in each sample, along with a taxa table recording taxonomy assigned to the ASVs from kingdom to species levels, using the naive Bayesian classifier algorithm and the UNITE database version 10.05.2021. Most ASVs had 251–336 bp, falling within the commonly amplified ITS1 region length of 200–600 bp. Next, we trimmed the ASV tables using the "phyloseq" package in R. ASVs with <10 sequences, i.e., rare ASVs, across all samples were removed. Before trimming, there were 22154 total ASVs

and 3118076 total sequences across 442 samples; afterwards, there were 15583 total ASVs and 3085446 total sequences. After removing rare ASVs, there were 4 to 126 ASVs (mean = 55) and 441 to 17234 sequences per sample (mean = 6981).

## Statistical analysis

For the lab incubation, we explored temporal trends in instantaneous C decomposition rate from each C source at each site and in lignin C decomposition rate for each individual sampling point (Supplementary Fig. 2), using GAMMs including an autoregressive error term to account for temporal autocorrelation, using the "mgcv" package[67] version 1.8.28 in R 3.6.1. Pairwise correlations between cumulative C decomposition over 6, 12, and 18 months (lignin, litter, soil and field lignin decomposition) and biogeochemical predictors were tested by Pearson correlation. The biogeochemical predictors included several categories, which we define as follows (1) climatic: MAT and MAP; (2) N-related: bulk N, bulk C/N, $NH_4^+$-N and $NO_3^-$-N after 1-, 9- and 18-month incubations; (3) geochemical: soil pH, silt+clay, $Al_{ox}$, $Fe_{ox}$, $Fe_{cd-ox}$, $Fe_{HCl}$, $Mn_{cd}$, $Ca_{cd}$; (4) microbial: fungal composition, fungal Chao1 richness, fungal quantity, bacterial quantity, and fungal-to-bacterial ratio (Supplementary Table 2).

In the microbial predictors, fungal composition was represented by the first (PC1) or second (PC2) axis of a principal coordinate analysis of ITS rRNA gene sequencing data on soils subsampled from the lab incubation at 14 months, conducted in the "vegan" package. The species-level abundance table (rather than the ASV table) was used to calculate Hellinger distances among samples before the analysis to alleviate the issue of a sparse matrix with many zero values[68]. The PC2 of fungal species composition was significantly ($P < 0.01$) correlated with cumulative lignin ($r = 0.37$) and litter ($r = 0.47$) decomposition in the lab incubation and was thus used as a fungal composition predictor. Similarly, the PC1 of fungal species composition was significantly ($P < 0.01$) correlated with cumulative SOC decomposition ($r = 0.35$). Overall fungal composition changed little with time during the lab incubation (Supplementary Fig. 5b). Therefore, for subsequent statistical analyses we used the ITS data from samples collected after 14 months of incubation, because only one sample from this timepoint was excluded from analyses because of low read counts. Fungal richness was represented by the residual of ASV Chao1 index regressed on the square root of the number of total sequences within a sample, a method that accounts for differences in sequencing depth among samples[69]. We used copy numbers of ITS and 16 S rRNA genes in the initial soil samples (1 g dry mass equivalent) as indices of fungal and bacterial quantity in our statistical models. Although fungal and bacterial quantities changed throughout the incubation (Supplementary Fig. 5a), including data from 9 and 14 months did not improve model performance. Fungal-to-bacterial ratio was calculated as fungal quantity divided by bacterial quantity.

We used LMMs and RFMs to identify important predictors for cumulative C decomposition (lignin, litter, soil, and field lignin) variables. We included the above-mentioned climatic, N-related, geochemical, and microbial predictors in models of the laboratory incubation decomposition data. Inorganic N predictors from three timepoints explained some variation in lab litter decomposition in the RFM but including these predictors did not improve model performance or change variable importance of other key predictors. Thus, inorganic N predictors were not retained in the final models, and we conducted the above-mentioned N addition experiment to specifically test the effects of inorganic N on lignin and litter decomposition. For statistical models of field lignin decomposition, we first fit the models including all categories of predictors and found that microbial predictors, silt+clay, and $Fe_{HCl}$ were not important predictors of field lignin decomposition. Therefore, we re-fit the models excluding these candidate predictors because these data were collected only for the field samples from the locations corresponding to the lab incubation.

Inorganic N variables in soil + litter mixtures were not measured for field lignin decomposition.

In the LMM, homoscedasticity and normality assumptions were met by raw data, except for lab lignin decomposition, which was log10 transformed. To estimate predictor importance, all variables were standardized to a mean of zero and a standard deviation of one to account for magnitude difference. All predictor variables were used as fixed effects and site was included as a random intercept to account for possible intra-site dependence in the LMMs. Adding sampling location as an additional random effect to account for correlations between 0–15 and 15–30 cm samples did not improve model performance. Some candidate predictors were excluded from initial models because of weak pairwise correlations with response variables (usually $r < 0.10$), and/or moderate-to-strong collinearities with other predictors (usually $r > 0.50$; Supplementary Table 5). We acknowledge that this approach might potentially exclude some important predictors that were correlated with other variables, but we found that decreasing the list of candidate predictor variables was important to achieve stable parameter estimates in cases of collinearity. Some predictors were further removed from final models through comparison of Akaike Information Criterion (AIC) values of nested models using stepwise backward selection. All predictors in the final models exhibited variance inflation factor values <3 and correlation coefficients <0.70 or >−0.70, implying that collinearity was acceptable. The relative contributions of fixed effects were determined by standardized regression coefficient estimates, and their significance was tested by the Wald chi-square test. The LMM performance was evaluated by $R^2$ representing variance explained by only the fixed effects and by the model, respectively. The LMM analyses were conducted with the "lme4" package[70].

We further used generalized additive mixed models (GAMMs) to verify the linearity of the important biogeochemical predictors defined in the LMMs. Details of RFM and GAMM analyses are described in the Supplementary Methods. All statistical analyses and plotting were performed in R statistical software version 3.6.1[66].

## Data availability
The data from this study[71] are available from the Environmental Data Initiative at https://doi.org/10.6073/pasta/3169668ed4727b41f8fbec1c0ebd46cb. The DNA sequencing data are available at National Center for Biotechnology Information (NCBI) Sequence Read Archive PRJNA808104 at https://www.ncbi.nlm.nih.gov/bioproject/PRJNA808104/. Source data are provided with this paper.

## Code availability
Code (R scripts) used to generate the figures and models are deposited to the Environmental Data Initiative at https://doi.org/10.6073/pasta/3169668ed4727b41f8fbec1c0ebd46cb.

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

## Acknowledgements
We thank all of the NEON staff who contributed to field sampling, and A. Mirabito, S. Tsui, L. James, A. Boyer, and H. Craven for lab assistance. We also thank Colin Williams for assistance with data visualization in Fig. 1. This work was funded in part by National Science Foundation grant 1802745 (S.J.H., S.R.W., A.H., C.L.). V.T. was partially funded by the U.S. Department of Energy Great Lakes Bioenergy Research Center (grants DE-FC02-07ER64494 and DE-SC0012742). The National Ecological Observatory Network is a program sponsored by the National Science Foundation and operated under a cooperative agreement by Battelle. Data collected/used in this research were obtained through the NEON Assignable Assets program.

## Author contributions

S.J.H. and S.R.W. conceived and designed this study. K.E.H. and V.I.T. conducted lignin syntheses. W.H. and W.Y. performed research. W.Y., E.R., and J.Y. conducted the microbial analysis. W.Y. and W.H. analyzed the data. W.H., W.Y., and S.J.H. wrote the manuscript. S.R.W., B.Y., K.E.H., C.L., and A.C.H. provided suggestions for substantial revisions. S.R.W. worked with the Senior Graphic Designer at NEON to generate Fig. 1 in Adobe Illustrator, using original graphics.

## Competing interests
The authors declare no competing interests.
