## [Peer Review File · Nature Communications]

Contrasting geochemical and fungal controls on decomposition of lignin and soil carbon at continental scaleReviewer #1 (Remarks to the Author):

In this manuscript the authors conduct a large, cross-site lignin/litter decomposition study, using both lab and in-field methods, over ~18 months. There is an interesting and important finding in the study – that their results on the differing control on lignin/litter and soil organic carbon (SOC) decomposition support the emerging view (and shifting paradigm) that lignin decomposition is not a “bottleneck” for SOC decomposition - i.e., that they are not linked, and so lignin may not (always) be a primary component of SOC. However, this is not very clear from the abstract and a bit buried in the current text where it is not highlighted until page 13 (the discussion). I suggest that this would be a more engaging framing for the manuscript than its current form. If re-written to highlight the novelty of the results and how the results shed light on the current hypotheses/paradigm surrounding connections among litter, lignin, and SOC decomposition and storage, it may be acceptable for publication.

The methods seem appropriate, although somewhat duplicative/repetitive. What does the linear mixed model provide that the random forest does not (or vice versa)? I am more familiar with boosted regression trees than random forest (RF), but I do like the lack of linearity assumptions that these analyses provide – especially given some of the strong nonlinearities in their RF results (see partial dependence plots in Figs. S8-11), I’m not certain that the linear model results are appropriate (e.g., the negative relationship of Feox with lab lignin decomposition which looks quite nonlinear in Fig. S8).

Finally, the paper (and discussion) is quite long. In general, I think focusing the manuscript on how the results inform SOC formation paradigms and focusing on one method (perhaps appending the other(s)?) would reduce the length and make the manuscript a stronger submission.

More specific comments and suggestions are below:

Abstract:

L21 alludes to the main point (i.e., lignin decomposition is not a “bottleneck” for SOC decomposition - i.e., that they are not linked, and lignin is not a primary component of SOC) but could be more clearly stated. Instead of focusing on accounting for variation, focus on the dominant paradigm; how do your results inform or change this? How do the authors think it will shift with climate or soil composition? The results definitely inform this.

Introduction:

L38: Suggest leading with this idea, e.g., “Traditionally, it was assumed that lignin, as an abundant biopolymer that protects components of plant tissue from microbial attack, limited both litter decomposition and thus contributed substantially to SOC. More recently, lignin’s importance in controlling litter and SOC decomposition has become controversial. Brief summary of recent work...

L46: The persistence of lignin may also vary systematically with climate (e.g., arctic vs anywhere warmer). Perhaps, e.g., “The persistence of lignin relative to SOC and other litter substrates might vary systematically with climatic, geochemical, and microbial characteristics, but the controls lignin, litter, and SOC decomposition have rarely been investigated together or across a wide range of climatic, geochemical or microbial variation.”

L51 paragraph: I suggest focusing on climate in this paragraph and N either in the next (soil characteristics) or its own. The authors have a very wide climate gradient, and the connections among lignin/litter decomposition and SOC decomposition (i.e., the contribution of lignin to SOC) may change across that gradient – it might be useful to consider this.

L94: If the authors can connect their work to the dominant paradigm here instead of “to test biogeochemical controls”, it would be more compelling.

L111-115: Again, how does testing these hypotheses inform the paradigm about decomposition and SOC?

Results - General comments

Sometimes it is hard to tell whether the results being discussed are from the lab incubation or the field incubation (e.g., beginning L177). Consistently using lab lignin decomposition vs field lignin decomposition would help or having individual paragraphs focused on each.

Does RF provide variable importance and the ability to examine interactions? Boosted regression trees (gbm and caret packages in R) address variable importance via relative influence and interactions via Friedman’s H-statistic (varies from 0 to 1 with higher values indicating larger interaction effects; package gbm) and so may be a preferable option. Particularly if the authors are interested in how decomposition drivers likely change across their large climate gradient.

L196-200: Ok, but some of the relationships are pretty nonlinear (e.g., for Feox and C/N). I would think this could affect the accuracy of the linear model results.

L234-5: I don’t see these variables in Fig. 5 for either analysis?

L235-7: How was a weak relationship defined? What were the variable effect sizes in general? Including the size of the increase/decrease would help readers understand variable importance (e.g., doubled, increased by X%).

Discussion

Lead with summary of how results changed/supported the paradigm.

In general, this discussion could be condensed by focusing on how the results changed or helped define the old/new paradigm.

L275-83: This seems like a really key takeaway. Leading with this would make the manuscript more compelling.

L324-28: It seems like the pre-statistical correlation analysis isn't needed since it doesn't account for covariates and is discounted here.

L408-10: The discussion of N availability impacts could be reduced to focus on this sentence.

L418-9: Seems late to introduce new acronyms and concepts. Overall, perhaps this discussion could be made more accessible by discussing controls on plant-particulate versus mineral associated organic carbon (or some more accessible terminology for general readers).

L435: a decrease in the ratio of POM/MAOM? So more POM less MAOM?

L450-2: Worth noting that this may not be true aboveground, especially in arid/high light environments subject to photodegradation.

Methods – see statistical comments above. Incubations and methods generally seem appropriate.

L533-5: Wouldn't CO₂-free air create diffusion of CO₂ out of the soil? Were jars tested for CO₂ saturation in the headspace?

L536: Was water added by weight to match field capacity?

L593: this is potential N mineralization, correct?

Fig S8-11. It would be useful to have rug plots along the x-axis of these plots that show the range and distribution of data (or some other way of showing this).

Fig S8. Label y axis as Lab lignin CO₂-C

Fig S9. Should this be CO₂? Or mass remaining/loss?

Reviewer #2 (Remarks to the Author):

This is a review for the Nature Communications submission «NCOMMS-22-38305” with the title “Contrasting geochemical and fungal controls on decomposition of lignin and soil carbon at continental scale” by Huang et al.

The manuscript deals with an important (and yet not well explored aspect) of organic matter decomposition in soil, namely the discrepancies between decomposition of different types of organic molecules with distinct quality compared to bulk SOC across large spatial soil gradients. The study has been performed across NEON Sites in the US, a very large environmental gradient and includes data that has been collected over several months during this extensive incubation experiment using isotopically labelled material.

I think the manuscript is an important contribution to our field, both because of the subject of the study and the spatial and temporal extent that it is covering. The authors have done a great job in putting these complex experimental data together and analyzed it with state of the art regression approaches. I would rank my comments on this manuscript to be something like a medium revision with no principal objection to any element of the presented study. But here are some suggestions that should be considered or commented on by the authors.

1. I think the interpretation of the many correlations explored in this dataset is sometimes getting a bit too far and speculative. While the authors try and generally succeed to back their interpretation

of certain correlations with literature findings, it still “just” the result of a regression. I would be more careful here and maybe shorten the discussion to some main points that you would like to take away. For example, calcium bridges have not been explored and many of the contrasting correlations between experimentally assessed variables (metal extracts) are a bit overinterpreted, but these are just examples. Since the bulk soils have not been fractionated to isolated functionally distinct soil C fractions I would be a bit careful with all the interpretation here.

2. Also, consider that the statistically approaches here will always favor some variables over others and “kick” the weak ones out to build a model that is less vulnerable to autocorrelation. This is good, but makes your remaining variables automatically to proxy variables that could stand for a much wider range of effects. The authors explain this a bit for the MAT and MAP effects, but the same is true for many other variables. Consider that these 20 Neon sites are extremely diverse in many, many ways.

3. I would suggest to try something like a rotated PCA to reduce the number of variables but see patterns in the potential predictors and how they group that allow you some sort interpretation beyond selecting single variables. If you do that, you will probably see some nice patterns in the predictors between the nutrient vs. metal soil variables, the climate vs. the C quality and so on. If you take that step, consider adding some interaction terms to the PCA so that you get more distinction in how to interpret your independent variables. For example MAP-PET, MAP:clay, clay:ph, clay:Fe and so on. Basically things that allow you to add more info on climate and soil properties that change mineralogy with effects on fertility and stabilization.

4. I was surprised to see not more discussion on the temporal development of carbon decomposition, compared to the quite lengthy and sometimes speculative regression analyses interpretation. If you follow my advice above and shrink/simplify this regression work, you could instead apply it also on distinct timepoints or time intervals (cumulative decomposition for the first 30day, 100days, end of the experiment and so on, whatever is meaningful). This should give some clarity on changing controls and the most influential group of factors throughout the experiment, which will certainly be different in the beginning compared to the middle or the end.

5. I am not really sure that the field incubation is that useful for this study. I am not convinced these results are really comparable to your very controlled conditions in the lab. The variability at a continental scale on weather and other relevant factors is just too large. And in fact in the discussion this takes space that you could save for other things. I would leave out the field experiment unless you can more clearly justify why this is important for this study and a clear way how you control for the variation in responses that you observe which relates to the variable conditions during the field trial at all these different places.

Minor comments:

- For me the strength of the NEON sites is not only that you get a lot of biogeochemical and climatic variation, but that you are dealing with very different pedogenetic stages and soil development

conditions that have shaped those explanatory variables you observe. This is almost not touched in this MS, but quite essential when interpreting the data. When hearing continental in the beginning of the manuscript, readers might think in plants and climate but you would need to make sure that your soil types are also described here. Since some of them will be quite representative for a climate zone and others be azonal soils that can have features that “trump” climate. Maybe elaborate this better in the methods and then pick up on it again in discussion to explain some of your observations?

- With respect to the last comment, when doing the lab experiment, what you call “legacy climate” [MAT/MAP] is no direct control and should not be treated as such in the analyses (this problem will go away if you do rotated PCA and see in which components they group). Over the long time of your incubation, the microbial community will fully adjust to those lab conditions and as such “climate” becomes irrelevant as a direct control. What they won’t adjust are the biogeochemical direct variation in your soils that have been shaped by climate.

- Why was Mnox not included? In near neutral pH soils, Mn should be an important electron acceptor and also binding partner for organic matter.

- I am not a native speaker but at two occasion it says “on one hand”. I think this should be “on the one hand”?

RESPONSE TO REVIEWERS' COMMENTS

Reviewer #1 (Remarks to the Author):

1. In this manuscript the authors conduct a large, cross-site lignin/litter decomposition study, using both lab and in-field methods, over ~18 months. There is an interesting and important finding in the study – that their results on the differing control on lignin/litter and soil organic carbon (SOC) decomposition support the emerging view (and shifting paradigm) that lignin decomposition is not a “bottleneck” for SOC decomposition - i.e., that they are not linked, and so lignin may not (always) be a primary component of SOC. However, this is not very clear from the abstract and a bit buried in the current text where it is not highlighted until page 13 (the discussion). I suggest that this would be a more engaging framing for the manuscript than its current form. If re-written to highlight the novelty of the results and how the results shed light on the current hypotheses/paradigm surrounding connections among litter, lignin, and SOC decomposition and storage, it may be acceptable for publication.

Response: Thank you for the reviewer’s positive comments on our manuscript. As suggested, we have re-written to highlight the novelty of the results by showing how the connections among litter, lignin, and SOC decomposition shed light on the classic and modern views of C decomposition. Please see our specific responses below.

2. The methods seem appropriate, although somewhat duplicative/repetitive. What does the linear mixed model provide that the random forest does not (or vice versa)? I am more familiar with boosted regression trees than random forest (RF), but I do like the lack of linearity assumptions that these analyses provide – especially given some of the strong nonlinearities in their RF results (see partial dependence plots in Figs. S8-11), I’m not certain that the linear model results are appropriate (e.g., the negative relationship of Feox with lab lignin decomposition which looks quite nonlinear in Fig. S8).

Response: This is a very useful comment and we appreciate the reviewer’s concerns about the linear mixed model (LMM) method, so we have conducted and presented additional analyses in the supplement of the revised manuscript to demonstrate the validity and parsimony of our approach. To reduce repetition, we have now focused on the LMM results in the main text and moved the other approaches to the supplement. We discuss boosted regression trees in detail below, and we emphasize that they are quite similar in their attributes to random forest model (RFM).

We used LMMs because they clearly reflect relationship strength and direction between predictors and the response variables, while RFMs could additionally explore nonlinear relationships if they are present. The RFM partial dependence plots showed that many relationships between predictors and response variables were approximately linear until predictors increased above the 90th or larger percentiles, where response variables became approximately constant (Supplementary Figs. 6–9). These partial dependence plots are consistent with an overall linear relationship between predictors and response variables, but to further test this assumption, we have also briefly included a third statistical method (generalized additive mixed models, GAMMs) in the revised MS. GAMMs are a variant of LMMs that allow for non-

linear relationships and fit a statistically optimum degree of curvature to the data, quantified using estimated degrees of freedom (edf), where a value of 1 is a linear relationship. Importantly, we found that almost all edf values were close to 1, indicating that a linear relationship was most parsimonious given the variability in the data (Supplemental Table 3). For the few predictors with degrees of freedom > 1, their relationships with C decomposition were similar to those observed in the RFMs (data not shown), i.e., the relationships were often approximately linear until predictors increased above the 90th or larger percentiles, where response variables became approximately constant (Supplementary Figs. 6–9). In this context, we should caution against over-interpreting the partial dependence plots of the RFMs in cases where the relationships are driven by only a few extreme points. Furthermore, as suggested below, we have added tick marks along the x-axis of these partial dependence plots to show the range and distribution of data (see Supplementary Figs. 6–9), which clearly indicated linear relationships for most of the data points.

Please see L169–176: “We used three statistical approaches (linear mixed models, LMMs, generalized additive mixed models, GAMMs, and random forest models, RFMs) to identify the most important soil geochemical, microbial, N, and climatic predictors of decomposition (Fig. 4). The RFM partial dependence plots showed that many relationships between predictors and response variables were approximately linear until predictors increased above the 90th or larger percentiles, where response variables became approximately constant (Supplementary Figs. 6–9). The GAMMs similarly demonstrated that nearly all predictors had linear relationships with response variables (Supplementary Table 3).”

Please see L683–685: “We further used generalized additive mixed models (GAMMs) to verify the linearity of the important biogeochemical predictors defined in the LMMs. Details of RFM and GAMM analyses are described in the Supplementary Methods.”

Please see Supplementary L135–142: “We further used generalized additive mixed models (GAMMs) to test whether the relationships among predictors and C decomposition variables were linear, using the “mgcv” package¹². GAMMs are a variant of LMMs that allow for non-linear relationships and fit a statistically optimum degree of curvature to the data. The degree is quantified using estimated degrees of freedom, where a value of 1 is a linear relationship and the higher the value, the more nonlinear the relationship is. We fit the GAMMs using the same predictors as in the LMMs, also accounting for the random effect of site and using a smooth function based on thin plate regression splines and an identity link function (default values in the mgcv package).”

Regarding the specific nonlinearity issue raised by the reviewer (Fe_{ox} and lab lignin decomposition), we note that lab lignin decomposition was not related to Fe_{ox} (Fig. 4 and Supplementary Fig. 6). We found that 90% of Fe_{ox} data were < 10 mg g⁻¹, and had an approximately linear relationship with field lignin decomposition in this range (Fig. 4 and Supplementary Fig. 7).

Thus, we have shortened our manuscript by moving the RFM results to the Supplementary Information and keeping the LMM results in the main text (Please see L169–208 in the main text for the LMM results and Supplementary L191–229 for the RFM results).

3. Finally, the paper (and discussion) is quite long. In general, I think focusing the manuscript on how the results inform SOC formation paradigms and focusing on one method (perhaps appendixing the other(s)?) would reduce the length and make the manuscript a stronger submission.

Response: This is another valuable comment. As suggested, we have trimmed text throughout the manuscript and have focused more on how the contradictory relationships of geochemical and microbial properties with lignin/litter vs. SOC decomposition inform SOC formation paradigms by reframing our discussion (L211–240). We also reduced the length of the manuscript by focusing on the LMM results (L169–208) and appending the RFM results in the Supplementary Information (Supplementary L191–229). Now the main text (excluding the method) has 4621 words (much fewer than 5532 words in our original version).

Based on discussions with colleagues (and feedback from peer reviewers on previous manuscripts), we are also aware of significant debate in the community about when the use of “paradigm” is merited. Rather than overstate the case regarding paradigms, which seems to raise ire among some scientists, we have attempted to address the spirit of this comment without using the word “paradigm” (we generally used “view” in these cases).

More specific comments and suggestions are below:

Abstract:

4. L21 alludes to the main point (i.e., lignin decomposition is not a “bottleneck” for SOC decomposition - i.e., that they are not linked, and lignin is not a primary component of SOC) but could be more clearly stated. Instead of focusing on accounting for variation, focus on the dominant paradigm; how do your results inform or change this? How do the authors think it will shift with climate or soil composition? The results definitely inform this.

Response: As suggested, we have revised the last sentence of the Abstract to clearly state the main point that lignin is not necessarily a bottleneck for SOC decomposition and can explain variable contributions of lignin to SOC among ecosystems (please see L31–33): “Decoupling of lignin and SOC decomposition and their contrasting biogeochemical drivers indicate that lignin is not necessarily a bottleneck for SOC decomposition and can explain variable contributions of lignin to SOC among ecosystems.”

This is a good point on how it might shift with climate or soil composition. As suggested, we have added a statement in the conclusion, although such information was not added to the Abstract due to the word limit stated by the journal (maximum 150 words). Please see L380–382: “Based on the observed differences in drivers of lignin decomposition vs. SOC decomposition, we might expect lignin to be a more important component of SOC in soils with higher Fe_{ox} and lower Mn_{cd} , Ca_{cd} and silt+clay.”

Introduction:

5. L38: *Suggest leading with this idea, e.g., “Traditionally, it was assumed that lignin, as an abundant biopolymer that protects components of plant tissue from microbial attack, limited both litter decomposition and thus contributed substantially to SOC. More recently, lignin’s importance in controlling litter and SOC decomposition has become controversial. Brief summary of recent work...”*

Response: As suggested, we have shortened these sentences by briefly summarizing recent work.

Please see L35–41: “Traditionally, it was assumed that lignin limits litter decomposition^{1,2} and contributes substantially to SOC^{3,4}. More recently, lignin’s importance in controlling litter and SOC decomposition has become controversial. Lignin might decompose fastest during early stages of litter decomposition^{5,6} and lignin-derived C could be less persistent in soil than other C components⁷. The contradictory views related to lignin decomposition and its contributions to SOC might be related to biogeochemical differences among ecosystems.”

6. L46: *The persistence of lignin may also vary systematically with climate (e.g., arctic vs anywhere warmer). Perhaps, e.g., “The persistence of lignin relative to SOC and other litter substrates might vary systematically with climatic, geochemical, and microbial characteristics, but the controls lignin, litter, and SOC decomposition have rarely been investigated together or across a wide range of climatic, geochemical or microbial variation.”*

Response: As suggested, we have revised these sentences to (L41–44) “The persistence of lignin relative to SOC and other litter components may vary systematically with climatic, geochemical and microbial characteristics across diverse soils⁸, but the controls on lignin, litter, and SOC decomposition have rarely been investigated together or across a wide range of climatic, geochemical or microbial variation.”

7. L51 paragraph: *I suggest focusing on climate in this paragraph and N either in the next (soil characteristics) or its own. The authors have a very wide climate gradient, and the connections among lignin/litter decomposition and SOC decomposition (i.e., the contribution of lignin to SOC) may change across that gradient – it might be useful to consider this.*

Response: As suggested, we have separated the climate and N factors. We first stated that the connection among lignin/litter decomposition and SOC decomposition may change across a wide climate gradient (L45–49) and then made the statements of N effects on lignin, litter and SOC decomposition as its own (L49–57).

Please see L45–49: “Climate can effectively predict litter decomposition at site to continental scales², but climate may affect decomposition of different C forms in different ways. For example, although high temperature and precipitation generally increase litter decomposition², they can also increase mineral weathering and C stabilization with reactive metals^{9,10}, which may specifically bind lignin-derived C^{11–13}.”

Please see L49–57: “In addition to climate, the ratio of lignin to nitrogen (N) is another conventionally important predictor of litter decomposition, but it may also have different

relationships with lignin, litter or SOC decomposition. Greater litter N content may increase lignin and litter decomposition by alleviating microbial N limitation^{1,14}, whereas increased N availability may also decrease decomposition of lignin or SOC by suppressing the production of oxidative enzymes⁴. Both mechanisms may occur in the same soils depending on the stage of litter decomposition; N may stimulate early stages while inhibiting later stages of litter decay^{14,15}. However, the overall importance of N relative to other soil characteristics remains poorly understood.”

8. L94: *If the authors can connect their work to the dominant paradigm here instead of “to test biogeochemical controls”, it would be more compelling.*

Response: We have revised this sentence by saying (L80–83) “To test competing viewpoints and potential mechanisms underlying the role of lignin in organic matter decomposition, we measured decomposition of lignin, bulk litter, and SOC via a uniform and quantitative isotopic method from mineral soil samples collected across broad biophysical gradients” so that we can connect our work to the dominant views of SOC.

9. L111-115: *Again, how does testing these hypotheses inform the paradigm about decomposition and SOC?*

Response: As also required by the journal to include a brief summary of the major results and conclusions of the current work, we have added a sentence to describe how testing the hypothesis inform the paradigm about decomposition and SOC (L105–108): “Our results support these hypotheses that partially reconcile aspects of classic and modern views of decomposition, such that lignin decomposition is a bottleneck for litter decomposition but not for SOC decomposition, thus explaining the variable contributions of lignin to SOC among soils as a function of their biogeochemical characteristics.”

Results - General comments

10. *Sometimes it is hard to tell whether the results being discussed are from the lab incubation or the field incubation (e.g., beginning L177). Consistently using lab lignin decomposition vs field lignin decomposition would help or having individual paragraphs focused on each.*

Response: Understood, we have now consistently used lab lignin decomposition vs. field lignin decomposition in our revised manuscript.

11. *Does RF provide variable importance and the ability to examine interactions? Boosted regression trees (gbm and caret packages in R) address variable importance via relative influence and interactions via Friedman’s H-statistic (varies from 0 to 1 with higher values indicating larger interaction effects; package gbm) and so may be a preferable option. Particularly if the authors are interested in how decomposition drivers likely change across their large climate gradient.*

Response: Excellent point. Yes, the RFM can provide variable importance as represented by IncMSE% in Fig. 4, and we can also examine interactions represented by Friedman’s H-statistic

(Supplementary Table 6). We have now explained these two indices in the Supplementary Methods section (Supplementary Methods L125–127 and L128–134).

Please see L125–127: “Variable importance was assessed using increase of mean squared error (%IncMSE) when a given variable is randomly permuted; a larger increase in MSE illustrates greater importance of the permuted variable.”

Please see L128–134: “To further explore variable interactions in the RFM, we first measured the overall interaction strength (H-statistic) for each variable in the model¹¹. The H-statistic is 0 if there is no interaction at all, and an H-statistic of 1 between two predictors means that each single PD function is constant and the effect on the prediction only comes through the interaction. Next, as we were particularly interested in how decomposition drivers might change across the large climate gradient, we calculated two-way interactions among MAP or MAT and other predictors and presented two-way PD plots with H-statistic > 0.05.”

It is worth noting here that boosted regression trees and RFM often exhibit very similar performance; for example, a recent paper found similar predicted distributions of SOC using boosted regression trees and RF (Yang et al., 2016, citation below), and RFM have the benefit of being less sensitive to tuning parameters. Thus, we prefer to keep the results from RFM analysis (as opposed to using yet another statistical approach).

According to the reviewer’s interesting suggestion, we have now examined the interactions of climate (MAT and MAP) with other biogeochemical properties. However, the overall interaction strengths of predictors were weak with H statistic < 0.25 (Supplementary Table 6). Relationships among biogeochemical properties and lab decomposition were similar across the climatic gradients. The findings are shown in Supplementary Fig. 12–14 and discussed in the Supplementary Results (Supplementary Results L219–228): “The overall interaction strengths of predictors were all < 0.25 (Supplementary Table 6) in lab decomposition RFMs, indicating relatively weak interactions. Despite generally higher lab lignin decomposition in soils with MAP > 1200 mm, relationships among lab lignin decomposition and predictors were often similar between < 1200 mm MAP and > 1200 mm MAP (Supplementary Fig. 12). Similarly, relationships among lab lignin decomposition and predictors were generally similar between < 3 °C MAT and > 3 °C MAT (Supplementary Fig. 12). For lab litter and SOC decomposition, the influence of other biogeochemical predictors was generally consistent across the climatic gradients (Supplementary Figs. 13 and 14). We did not explore interactions in the field lignin RFM given the absence of some predictors that were included in the lab lignin RFM.”

Yang, R. M. *et al.* Comparison of boosted regression tree and random forest models for mapping topsoil organic carbon concentration in an alpine ecosystem. *Ecol. Indic.* **60**, 870–878 (2016).

12. L196-200: *Ok, but some of the relationships are pretty nonlinear (e.g., for Feox and C/N). I would think this could affect the accuracy of the linear model results.*

Response: We appreciate the reviewer’s concern on the apparent nonlinear relationships from the partial dependence plots, which are driven mostly by a few points with strong leverage as described above. To demonstrate the accuracy of the linear mixed model results, as described above we further used the generalized additive mixed models (GAMMs) that account for possible nonlinear relationships between predictors and responses. The results from the GAMMs showed that the defined important predictors (e.g. Fe_{ox} and C/N) had linear relationships with the lab lignin decomposition, except for the fungal quantity (Supplementary Table 3). In the manuscript, we have been careful with cases of stronger nonlinearity in the Supplementary Notes; for example, we did not specifically discuss the relationship between C/N and lab lignin decomposition which was nonlinear as shown in the RFM (Supplementary Fig. 6).

13. L234-5: I don’t see these variables in Fig. 5 for either analysis?

Response: Sorry for the confusion. We have revised the sentence to make it clear by stating (Supplementary Results L208–212): “The optimal RFM of lab litter decomposition explained 60% of the variation, and included all of the 17 biogeochemical predictors except for silt+clay and soil C/N. The predictors shared with the LMM all had directionally similar correlations with litter decomposition, illustrated by comparing model coefficients (Fig. 4) and the RFM partial dependence plot (Supplementary Fig. 8).”

14. L235-7: Howe was a weak relationship defined? What were the variable effect sizes in general? Including the size of the increase/decrease would help readers understand variable importance (e.g., doubled, increased by X%).

Response: Good point, we have deleted “weak” here, and quantified the effect size using the IncMSE. The key point here is the negative relationships of lab litter decomposition with Al_{ox}, Fe_{ox}, and pH (Supplementary Results L212–213). All these predictors had a IncMSE% > 10%.

Please see Supplementary Results L212–213: “Fe_{ox}, Al_{ox}, and pH had negative relationships with litter decomposition in the RFM while most other predictors had positive relationships (Supplementary Fig. 8).”

Discussion

15. Lead with summary of how results changed/supported the paradigm.

Response: As suggested below, we have now led this section with the summary of how our results supported aspects of classic vs. modern views of C decomposition.

Please see L211–230: “Overall, our continental-scale data showed the particular importance of geochemical and microbial predictors for lignin and litter decomposition, and their differing relationships with SOC decomposition (Fig. 4), consistent with our first and second hypothesis, respectively. Our results collectively supported different aspects of classic and modern views of decomposition. The strong correlation between lignin and litter decomposition (Fig. 2) and the similar biogeochemical predictors of these processes support the classic view that lignin decomposition is tightly coupled with overall litter decomposition^{1,27}. However, we found that

decomposition of SOC was unrelated to decomposition of lignin and litter, and that these processes often had contrasting relationships with biogeochemical predictors. Several soil geochemical factors had negative (Fe_{ox} and Al_{ox}) and positive (Fe_{HCl} , Mn_{cd} and Ca_{cd}) relationships with lignin and/or litter decomposition, while there were almost entirely negative relationships between extractable soil metals and SOC decomposition (Fig. 4; Supplementary Fig. 10). Intriguingly, microbial variables including fungal composition and fungal and bacterial quantity were needed to explain variation in the decomposition of lignin and litter, but not SOC (Fig. 4). These findings are inconsistent with the classic idea that the slow decomposition of lignin residues limits decomposition of total SOC³. Rather, the disparate rates and predictors of lignin and SOC decomposition support the modern proposal that lignin depolymerization is not necessarily a primary bottleneck for SOC decomposition^{5,7}. Furthermore, our dataset provides an explanation for the decoupling of lignin and SOC decomposition by highlighting their differing relationships with geochemical and microbial variables.”

16. In general, this discussion could be condensed by focusing on how the results changed or helped define the old/new paradigm.

Response: As suggested, we have reduced the length of our manuscript by focusing on how results helped define the classic vs. modern views of C decomposition. We also have shortened the discussion on the effect of N availability on C decomposition. Please see our specific response below.

17. L275-83: This seems like a really key takeaway. Leading with this would make the manuscript more compelling.

Response: As suggested, we have moved this information to the beginning of the discussion. Please see L211–240.

18. L324-28: It seems like the pre-statistical correlation analysis isn't needed since it doesn't account for covariates and is discounted here.

Response: Good point, we have moved the pairwise correlations to the Supplementary Information (Supplementary Fig. 10). We kept the results of pairwise relationships in the Supplementary Information (Supplementary Notes L154–189) as the results were straightforward and may be interesting to some readers.

19. L408-10: The discussion of N availability impacts could be reduced to focus on this sentence.

Response: Understood, we have reduced the discussion on N availability impacts and focused on the main point of relatively less importance of N availability on lignin and litter decomposition than geochemical and microbial predictors.

Please see L328–336: “On balance, inorganic N addition led to only a small net stimulation of lab lignin and litter decomposition after 18 months (Fig. 3), and the effects of N were relatively small in comparison with variation across sites (Figs. 2 and 3). The positive response of lignin and litter decomposition to N addition might imply that microbial growth was N-limited in many

sites. However, lab lignin and litter decomposition were not consistently related to inorganic N in the experiment without N addition, and they had differing relationships with total N and C:N (Fig. 4, Supplementary Figs. 6, 8 and 10). When considering continental-scale variation in biogeochemical properties, variation in N availability may be a less important driver of decomposition than sometimes assumed.”

20. L418-9: Seems late to introduce new acronyms and concepts. Overall, perhaps this discussion could be made more accessible by discussing controls on plant-particulate versus mineral associated organic carbon (or some more accessible terminology for general readers).

Response: As suggested, we have used “particulate organic C” and “mineral associated organic C” and defined the likely sources of these C fractions (particulate organic C likely derived mostly from plants, and mineral associated organic C likely derived from a mixture of plants and microbes) to make them more accessible for general readers in this paragraph (L341–343).

Please see L341–343: “...defined as chemically dispersed particulate organic C (> 53 μm , likely derived mostly from plants) and mineral-associated organic C (< 53 μm , likely derived from a mixture of plants and microbes), ...”

21. L435: a decrease in the ratio of POM/MAOM? So more POM less MAOM?

Response: Sorry for the confusion. We have revised the sentences to make it clear (L358–363): “Similarly, the positive relationship between silt+clay and lignin decomposition and its negative relationship with SOC decomposition is consistent with our previous finding that increased silt+clay was associated with lower particulate vs. mineral-associated organic C⁵³. This might simply be due to increased capacity for mineral protection, but it might also be linked to increased catalysis of lignin decomposition by metals and/or minerals in these fine particle fractions³⁸.”

22. L450-2: Worth noting that this may not be true aboveground, especially in arid/high light environments subject to photodegradation.

Response: Agreed. We have stated “this might differ in arid or high light environments subject to photodegradation⁵⁶” (L377–378) and also cited Adair et al., 2008 to support this statement.

23. Methods – see statistical comments above. Incubations and methods generally seem appropriate.

Response: For the comments regarding the statistical analysis, we have now explained our rationale for focusing on the LMM in the main text (L169–176). We have also added the interactions of climate with other important biogeochemical predictors in the RFMs (Supplementary Methods L219–228; Supplementary Figs. 12–14).

24. L533-5: Wouldn't CO₂-free air create diffusion of CO₂ out of the soil? Were jars tested for CO₂ saturation in the headspace?

Response: Yes, this is the rationale of our incubation design. Because the volume of incubated soil was ~1000-fold smaller than the jar headspace, CO₂ produced by soil microbes would diffuse out of the soil and accumulate in the headspace with negligible storage in soil pores. By periodically flushing the jars, CO₂ concentrations remained below 5000 ppm during the incubation. Assuming a 1:1 ratio between CO₂ production and oxygen (O₂) consumption, O₂ decreased by < 2.4% of the initial value (20.9%) during each sampling period. We have added this information in the Methods.

Please see L474–480: “The jars were flushed with CO₂-free air following periodic headspace sampling as described below, and CO₂ concentrations remained below 5000 ppm during the incubation. Assuming a 1:1 ratio between CO₂ production and oxygen (O₂) consumption, O₂ decreased by < 2.4% of the initial value (20.9%) during each sampling period. Because the volume of incubated soil was ~1000-fold smaller than the jar headspace, CO₂ produced by soil microbes would diffuse out of the soil and accumulate in the headspace with negligible storage in soil pores.”

25. L536: *Was water added by weight to match field capacity?*

Response: Yes, soil moisture was monitored by recording the mass of each sample, and water was added as necessary to match the original mass of each sample under field moisture capacity every month before 179 d and every other month thereafter (due to the less frequent gas sampling) to replenish vapor lost during headspace flushing. We have revised the sentence to make it clear.

Please see L480–484: “Soil moisture was monitored by recording the mass of each sample, and water was added as necessary to match the original mass of each sample under field moisture capacity every month before 179 d and every other month thereafter (due to the less frequent gas sampling) to replenish vapor lost during headspace flushing.”

26. L593: *this is potential N mineralization, correct?*

Response: Yes, we measured the net change in mineral N between time periods, which is frequently defined as net N mineralization, but this measure is also sometimes referred to as “Potential N mineralization” (L539–540), so we have included this terminology as well (e.g. <https://lter.kbs.msu.edu/protocols/34>).

Please see L539–540: “Net N mineralization (sometimes known as potential N mineralization) was calculated as...”

27. Fig S8-11. *It would be useful to have rug plots along the x-axis of these plots that show the range and distribution of data (or some other way of showing this).*

Response: As suggested, we have added tick marks along the x-axes of these plots to show the range and distribution of data (Supplementary Figs. 6–9). For the newly added RFM interaction plots, we also added tick marks along both x- and y-axes (Supplementary Figs. 12–14).

28. *Fig S8. Label y axis as Lab lignin CO₂-C*

Response: Agreed. We have changed the y-axis label to “Lab lignin CO₂-C”. Please see Supplementary Fig. 6. We have also changed the y-axis labels in Supplementary Figs. 8 and 9 to “Lab litter CO₂-C” and “Lab SOC CO₂-C”, respectively.

29. *Fig S9. Should this be CO₂? Or mass remaining/loss?*

Response: As suggested, we have changed the y-axis label to “Field lignin C loss” in Supplementary Fig. 7.

Reviewer #2 (Remarks to the Author):

This is a review for the Nature Communications submission «NCOMMS-22-38305” with the title “Contrasting geochemical and fungal controls on decomposition of lignin and soil carbon at continental scale” by Huang et al.

The manuscript deals with an important (and yet not well explored aspect) of organic matter decomposition in soil, namely the discrepancies between decomposition of different types of organic molecules with distinct quality compared to bulk SOC across large spatial soil gradients. The study has been performed across NEON Sites in the US, a very large environmental gradient and includes data that has been collected over several months during this extensive incubation experiment using isotopically labelled material.

I think the manuscript is an important contribution to our field, both because of the subject of the study and the spatial and temporal extent that it is covering. The authors have done a great job in putting these complex experimental data together and analyzed it with state of the art regression approaches. I would rank my comments on this manuscript to be something like a medium revision with no principal objection to any element of the presented study. But here are some suggestions that should be considered or commented on by the authors.

Response: We appreciate the reviewer’s interest in this work and for their thoughtful comments.

1. I think the interpretation of the many correlations explored in this dataset is sometimes getting a bit too far and speculative. While the authors try and generally succeed to back their interpretation of certain correlations with literature findings, it still “just” the result of a regression. I would be more careful here and maybe shorten the discussion to some main points that you would like to take away. For example, calcium bridges have not been explored and many of the contrasting correlations between experimentally assessed variables (metal extracts) are a bit overinterpreted, but these are just examples. Since the bulk soils have not been fractionated to isolated functionally distinct soil C fractions I would be a bit careful with all the interpretation here.

Response: We appreciate this point of view. As suggested, we have deleted several sentences on the interpretation of soil metal relationships with C decomposition throughout the paper, and made the argument more tentative in the new text (L258–268): “The finding that Fe_{HCl} was positively related and Fe_{ox} was negatively related with lignin and litter decomposition was consistent with multiple functional roles of Fe, which might stimulate decomposition or provide protection depending on C molecular composition and/or redox environment^{13,34}. Moreover, lignin and litter decomposition increased in samples with greater Mn and Ca (Fig. 4), consistent with the importance of Mn-promoted degradation of organic C¹⁸. Mn can promote lignin decomposition via enzymes and redox cycling³⁵, which may have increased overall litter decomposition¹⁹. The strong positive relationships between Ca and decomposition of lignin and litter agreed with previous studies showing that Ca was positively related to the extent of litter mass loss, and in particular, lignin degradation^{21,36} as Ca is an essential component of the fungal cell wall and can increase the growth of white rot fungi³⁷.”

In response to this comment we have also shortened our discussion and focused more on the main points of how the contradictory relationships of geochemical and microbial properties with lignin/litter vs. SOC decomposition informed the classic and modern views of C decomposition, as suggested by the first reviewer. Please see L211–240.

2. Also, consider that the statistically approaches here will always favor some variables over others and “kick” the weak ones out to build a model that is less vulnerable to autocorrelation. This is good, but makes your remaining variables automatically to proxy variables that could stand for a much wider range of effects. The authors explain this a bit for the MAT and MAP effects, but the same is true for many other variables. Consider that these 20 Neon sites are extremely diverse in many, many ways.

Response: This is an interesting point, and we have now included further description and rationale for our statistical approach. We emphasize that our final LMM and RFM models (Fig. 4) actually included a large subset of our candidate predictor variables shown in Supplementary Figure 1. That is, many of the potential biogeochemical predictors ended up being retained in the optimum models selected by AIC, even after addressing issues of autocorrelation, which was critical to obtain models with stable parameters. We have noted this on L177–178: “Most of our optimal statistical models included multiple predictor variables from most categories (N-related, geochemical, and microbial variables; Fig. 4).”

We have now also addressed this point in the Methods (L672–675): “We acknowledge that this approach might potentially exclude some important predictors that were correlated with other variables, but we found that decreasing the list of candidate predictor variables was important to achieve stable parameter estimates in cases of collinearity.”

For clarity, we have presented the full list of initial candidate variables used to select each model, after considering autocorrelation, in Supplementary Table 5.

3. I would suggest to try something like a rotated PCA to reduce the number of variables but see patterns in the potential predictors and how they group that allow you some sort interpretation beyond selecting single variables. If you do that, you will probably see some nice patterns in the predictors between the nutrient vs. metal soil variables, the climate vs. the C quality and so on. If you take that step, consider adding some interaction terms to the PCA so that you get more distinction in how to interpret your independent variables. For example MAP-PET, MAP:clay, clay:ph, clay:Fe and so on. Basically things that allow you to add more info on climate and soil properties that change mineralogy with effects on fertility and stabilization.

Response: We thank the reviewer for suggesting the PCA method, although the following argument and results demonstrate that this method may not be as appropriate as our original methods in serving our goals in this study. We have now included an analysis of climate interactions in the RFM, and we showed that these interactions were minor. Please see the supplement (Supplementary Results L219–228; Supplementary Figs. 12–14), and also the response to Reviewer 1.

First, we sought to compare relationships among individual biogeochemical predictors and C decomposition, rather than groups of predictors, because we expected that different predictors (e.g. metals) would have categorically different relationships with our response variables (e.g., lignin and SOC decomposition). For example, both the LMM and the RFM showed that Mn_{ox} was positively related to lab litter decomposition whereas Fe_{ox} was negatively related to lab litter decomposition (Fig. 4). These relationships are consistent with fundamentally different biogeochemical mechanisms demonstrated in previous studies (a catalytic role for Mn in decomposition and a protective role for Fe), as discussed in the main text (L258–261 and L263–264). While PCA has advantages in reducing the dimensionality of the predictors, by lumping multiple predictors together in PCA axes, we eliminate the contrasting relationships between predictors and response variables that we would expect from theory.

To illustrate this point, we compared the results of PCA and LMM. First, we extracted the first principal component (PC1) from the categories of climatic (2), N-related (2), geochemical (8), and microbial (5) predictors, respectively, as defined in the main text (L151–153 and L650–655). These PCs could represent 38.7%, 32%, 67.5%, and 53.7% of the overall variation in each predictor category, respectively. We then used these PCs along with their two-way interactions in new LMMs to identify the overall importance of these categories. These PCs, even along with their interactions, explained a much smaller proportion of the variation in lab decomposition (R^2_{fixed} from 0.26–0.35 in Response Table 1) as compared to individual predictors (R^2_{fixed} from 0.43–0.49 in Fig. 4). Moreover, some predictors which we would expect to be important according to theory became insignificant in the LMM based on PCA scores. For example, microbial PC1 was insignificant for lab lignin decomposition in the PCA-based model, in contrast to our original model that separately included the composition and quantity of fungal communities (Fig. 4). For completeness, we report the results from these analyses in the table below, but we chose not to include them in the revised Supplement as there are already two other additional statistical approaches reported there (the RFM and GAMM, as described in the response to reviewer 1 above).

Reviewer Response Table 1 Effects (standardized coefficients) of predictor PC1 on cumulative lignin, litter, and SOC decomposition after 18 months of lab incubation in LMMs

Predictors	Lab lignin decomposition	Lab litter decomposition	Lab SOC decomposition
R-squared	$R^2_{fixed} = 0.255$ $R^2_{fixed + random} = 0.488$	$R^2_{fixed} = 0.348$ $R^2_{fixed + random} = 0.482$	$R^2_{fixed} = 0.338$ $R^2_{fixed + random} = 0.755$
Microbial PC1	0.14	0.41*	-0.19
Geochemical PC1	-0.27*	-0.09	0.19
Climatic PC1	0.31*	0.19	0.30
N-related PC1	-0.16	-0.40*	-0.70*
Microbial PC1 × Geochemical PC1	0.26*	0.11	-0.15
Microbial PC1 × Climatic PC1	-0.01	-0.04	-0.00
Microbial PC1 × N-related PC1	-0.03	0.03	-0.21

Geochemical PC1 × Climatic PC1	-0.14	0.08	0.15*
Geochemical PC1 × N-related PC1	-0.00	0.06	-0.26*
Climatic PC1 × N-related PC1	0.13	0.19	0.37*

* denotes $P < 0.05$.

Another approach to use PCA for regression that does not rely on a-priori grouping of variables is principal component regression (PCR), and we have discussed this method with a faculty colleague in the Statistics Department at Iowa State University (Prof. Somak Dutta, Personal Communication, Dec 22, 2022). The PCR involves constructing several principal components (PCs), i.e., new uncorrelated variables that successively maximize variance, and then using these components as the predictors in a linear regression model that is fit using least squares. However, because it is used for feature generation rather than feature selection, PCR is used to create a prediction model but not a model with mechanistically interpretable coefficients (James et al. 2013). Despite the interpretability issue, we still explored PCR using the following three sets of predictors: 1) the 17 predictors on climatic, N-related, geochemical, and microbial characteristics; 2) the 17 predictors and 30 interactions among climatic and other predictors (15 for MAP and 15 for MAT); and 3) the 17 predictors and 30 interactions as well as 28 interactions among geochemical predictors. The first principal component (PC1) could only explain 14.7%, 16.1%, and 15.0% of the variance in lab lignin decomposition in these three models, which was much lower compared with directly using the predictors (45% and 51% for LMM and RFM, respectively, in Fig. 4). Thus, further identifying predictor importance based on loading value of each predictor on PC1 seems not very reliable for this dataset, and the PC1 in PCR of other C decomposition variables was similarly low.

Thus, we argue that PCA and PCR may not be appropriate methods for exploring the relationships among biogeochemical predictors and C decomposition in this study.

James, G., Witten, D., Hastie, T., & Tibshirani, R. 2013. *An Introduction to Statistical Learning*. Edited by G. Casella, Fienberg S, and I. Olkin. New York: Springer.

4. I was surprised to see not more discussion on the temporal development of carbon decomposition, compared to the quite lengthy and sometimes speculative regression analyses interpretation. If you follow my advice above and shrink/simplify this regression work, you could instead apply it also on distinct timepoints or time intervals (cumulative decomposition for the first 30day, 100days, end of the experiment and so on, whatever is meaningful). This should give some clarity on changing controls and the most influential group of factors throughout the experiment, which will certainly be different in the beginning compared to the middle or the end.

Response: Thank you for your suggestion. We emphasize here that we already conducted a similar analysis in the initial Results section, which is now presented in the Supplementary Results to address Reviewer 1's concerns about the need to concisely present our main results. Briefly, we assessed relationships among predictors and lab decomposition after 6, 12 and 18 months of cumulative decomposition using both pairwise correlation and LMM (Supplementary

Figs. 10 and 11). Importantly, the importance of these predictors changed little over time, i.e., the most influential group of factors was relatively consistent in the beginning compared to the middle or the end of the experiment. Thus, we focused our subsequent analysis on the 18-month (571 d) dataset. Given that PCA had poorer performance and could not fully meet our demand of exploring nuances of relationships as compared with the LMM or RFM in this study, we did not perform PCA for 6- and 12-month incubation data. We describe the logic for focusing on the 18-month data on L180–184: “In addition, to test whether biogeochemical predictors changed with time throughout lab incubation, we compared models of cumulative decomposition after 6, 12, and 18 months (Supplementary Figs. 10 and 11). Predictors were generally similar over time, and thus we focused our subsequent analysis on the 18-month (571 d) dataset (the other results are presented in the Supplemental Information).”

5. I am not really sure that the field incubation is that useful for this study. I am not convinced these results are really comparable to your very controlled conditions in the lab. The variability at a continental scale on weather and other relevant factors is just too large. And in fact in the discussion this takes space that you could save for other things. I would leave out the field experiment unless you can more clearly justify why this is important for this study and a clear way how you control for the variation in responses that you observe which relates to the variable conditions during the field trial at all these different places.

Response: We appreciate this point, and in the revised text we have now explicitly provided our rationale for including the field incubation data (L138–142 and L399–404). We suggest that the field experiment data is a strength of our overall study that bolsters the confidence in our laboratory measurements, despite the fact that they were conducted under standardized temperature/moisture conditions that were sometimes very different from the site mean values. This is illustrated by the fact that the predictors in our optimum statistical models of lab and field lignin decomposition were generally similar, despite the differences in environmental conditions between field and lab. In fact, incorporating the real variability in temperature and moisture among the sites, as well as providing an opportunity for additional microbial colonization of soil/litter/lignin substrates, was the goal of the field study.

Please see text on L138–142: “Our field sites had large climate differences whereas the lab samples were incubated at the same temperature and comparable moisture, so we used an additional field lignin decomposition experiment with 0–15 cm soils to test whether similar biogeochemical predictors were important in the field and in the lab. The field experiment was conducted in mesh bags that allowed additional microbes to colonize the soil/litter/lignin mixtures over time.”

Please see text on L399–404: “The results of lignin decomposition and its predictors from the lab incubation were further compared with those from a field incubation. The lab incubation enabled us to compare C decomposition among samples while standardizing temperature and moisture, whereas the field incubation allowed us to assess effects of actual site temperature and moisture on lignin decomposition, while also allowing for sample colonization by additional microbes.”

We are not certain what the reviewer is referring to when they ask us to control for the variation in environmental conditions among sites. Rather, exposing the soil/litter/lignin substrates to a

very broad range of field temperature and moisture conditions (tundra to tropics), and then assessing their impacts on decomposition, were the goals of the study. Indeed, we found that actual site temperature and moisture were important predictors of lignin in the field, and that the legacy of those long-term differences was important in terms of predicting lab lignin decomposition.

Please see text on L311–325: “After accounting for other biogeochemical predictors, MAT and MAP of the study sites was still related to organic matter decomposition (Fig. 4) even under the common conditions of temperature and moisture imposed in the lab incubation. This is consistent with previous findings that climate history influenced litter and SOC decomposition, possibly by shaping the composition and functional responses of decomposer communities and/or via correlation with soil minerals^{10,51,52}. Microbial communities from different soils can remain distinct over months to years even when exposed to a common temperature and moisture regime, and in spite of changes in community composition over time^{45,46} (e.g. Supplemental Fig. 5). Climate greatly impacts soil weathering^{9,10}, and although our statistical models included geochemical variables, it is also possible that the apparent relationships between decomposition and climate also reflected geochemical differences that were not accounted for by the extractable metals data (Fig. 4). It was not surprising that MAP had different relationships with field and lab lignin decomposition (Fig. 4, Supplementary Figs. 6 and 7), given that MAP reflected either the actual differences in climate during the field experiment or the legacies of prior differences in climate during the lab experiment.”

Minor comments:

6. For me the strength of the NEON sites is not only that you get a lot of biogeochemical and climatic variation, but that you are dealing with very different pedogenetic stages and soil development conditions that have shaped those explanatory variables you observe. This is almost not touched in this MS, but quite essential when interpreting the data. When hearing continental in the beginning of the manuscript, readers might think in plants and climate but you would need to make sure that your soil types are also described here. Since some of them will be quite representative for a climate zone and others be azonal soils that can have features that “trump” climate. Maybe elaborate this better in the methods and then pick up on it again in discussion to explain some of your observations?

Response: This is a good point. As suggested, we have added more information to describe soil types in the Methods (L418–421) and discuss some specific soils in the Results (L124–128).

Please see L418–421: “The nine soil orders include Alfisols (CPER and SCBI), Andisols (WREF), Aridisols (ONAQ and SRER), Entisols (DSNY, OSBS and PUUM), Gelisols (TOOL), Inceptisols (BONA, GRSM, HARV, LENO and NIWO), Mollisols (KONZ, SJER, WOOD and YELL), Spodosols (UNDE), and Ultisols (TALL).”

Please see L124–128: “The soils with coolest climate (TOOL, a Gelisol) showed the lowest site-averaged decomposition of lab lignin (3.1%), litter (15.4%) and SOC (13.3%) relative to other soils. The soils with warm and dry climate (ONAQ and SRER, Aridisols) had relatively lower site-averaged lab lignin (4.1%) and litter C (19.6%) decomposition but the highest site-averaged SOC decomposition (52.0%).”

For further clarity we have provided additional detail regarding the NEON site selection design, which is based on climate, not soil characteristics. As a result, these sites had very large differences in soil properties, but the soils cannot necessarily be viewed as representative of particular climate regions. Rather, the soils were chosen to span a very large range of biogeochemical characteristics that are thought to influence OM decomposition. In this context, we are not making a claim as to whether particular NEON soils are representative or azonal for a region, but simply using the diversity of soil characteristics across these sites to explore potential drivers of OM decomposition. We have also clarified the language in the paper to indicate that our sampling was not representative of North America as a whole, even though the soils were collected from sites across North America, which was our rationale for using the phrase “continental scale” (see L24–26 in the Abstract, L83–88 and L408–416).

Please see L24–26: “Here we tracked decomposition of a lignin/litter mixture and SOC across different North American mineral soils using lab and field incubations.”

Please see L83–88: “We used 20 sites from the US National Ecological Observatory Network (NEON) that span diverse ecosystems and climatic zones (tundra to tropics). These particular samples were not necessarily representative of North American soils as a whole, but were instead selected to span a broad range of biogeochemical properties thought to influence C decomposition and accrual; they included 9 of 12 orders in the USDA soil taxonomy (Fig. 1a, b and c, Supplementary Fig. 1 and Table 1).”

Please see L408–416: “NEON sites are stratified among domains defined by climate characteristics⁵⁸, not by soil type, and while they naturally contain a wide diversity of soil types, soils at each site are not necessarily representative of the corresponding ecoclimatic domain. For this project, we selected 20 NEON terrestrial sites, denoted by their acronyms as follows: BONA, CPER, DSNY, GRSM, HARV, KONZ, LENO, NIWO, ONAQ, OSBS, PUUM, SJER, SRER, SCBI, TALL, TOOL, UNDE, WREF, WOOD, YELL (Fig. 1a). These sites span wide edaphic, climatic and ecosystem gradients (Supplementary Fig. 1), and they were chosen to span broad differences in biogeochemical characteristics, within constraints of feasibility and permitting.”

7. With respect to the last comment, when doing the lab experiment, what you call “legacy climate” [MAT/MAP] is no direct control and should not be treated as such in the analyses (this problem will go away if you do rotated PCA and see in which components they group). Over the long time of your incubation, the microbial community will fully adjust to those lab conditions and as such “climate” becomes irrelevant as a direct control. What they won’t adjust are the biogeochemical direct variation in your soils that have been shaped by climate.

Response: Please see the above discussion regarding the issues with using PCA-derived composite variables in our statistical models. The reviewer has brought up an interesting question of whether the legacy effect of climate on soil microbial communities was still present after 18 months. There is clear prior evidence of legacy effects of climate on soil microbial communities when incubated under common moisture and temperature treatments, and the time span that those effects could last varied between 3 to 18 months according to the prior

experiments (Bradford et al., 2019; Glassman et al., 2018; Wang and Allison, 2022). It is correct that microbial communities also change over time during incubations, yet initially different communities maintain their differences in spite of temporal variation (Strickland et al. 2009, Cleveland et al. 2014, cited in the main manuscript). Thus, these previous studies clearly demonstrate that climate could exert a legacy effect on OM decomposition through differences in microbial communities. Another possible explanation for apparent effect of legacy climate on decomposition involves the effects of climate on soil weathering; we included geochemical variables (extractable metals) in our models, but these may not have entirely accounted geochemical changes linked to climate, so we cannot conclusively determine the mechanism by which legacy climate variables related to decomposition in our study.

However, irrespective of the precise interpretation, the legacy climate variables (MAT and in some cases MAP) were clearly important in our models; for example, removing them increased model AIC in all cases, sometimes by as much as 5 points (compare an AIC value of 373.3 for the optimal LMM of lab lignin decomposition, which included MAT, vs. an AIC value of 378.8 for the model without MAT).

We have included discussion of these issues and references (Strickland et al. 2009; Cleveland et al. 2014; Glassman et al., 2018) supporting climate legacy effects on microbes and geochemistry in the text (L311–325).

Additional references:

Bradford, M. A., *et al.* Cross-biome patterns in soil microbial respiration predictable from evolutionary theory on thermal adaptation. *Nat. Ecol. Evol.* **3**, 223–231 (2019).

Wang, B., & Allison, S.D. Climate-driven legacies in simulated microbial communities alter litter decomposition rates. *Front. Ecol. Evol.* 10:841824 (2022).

8. *Why was Mn_{ox} not included? In near neutral pH soils, Mn should be an important electron acceptor and also binding partner for organic matter.*

Response: We also measured Mn_{ox} in our samples. However, the Mn_{ox} had very similar values with the Mn_{cd} ($r = 0.98$). Thus, we only reported Mn_{cd} in our manuscript. We previously stated this in Methods and now added the value of correlation coefficient. Please see L561–563: “Extractions of Al and Mn by oxalate and citrate-dithionite were very similar ($r = 0.88$ and $P < 0.001$ for Al; $r = 0.98$ and $P < 0.001$ for Mn), so we only report Al_{ox} and Mn_{cd}.”

9. I am not a native speaker but at two occasion it says “on one hand”. I think this should be “on the one hand”?

Response: Both “on one hand” and “on the one had” are correct uses of the phrase, but we have deleted it in an effort to reduce length, per the suggestion of Reviewer 1.

Reviewer #2 (Remarks to the Author):

Overall, the authors did a great and very thorough job of addressing the reviews. The revised manuscript is very clear and more compelling. Excellent work. One minor comment is that I would suggest adding "Lab" to L99 to make it clear that the N addition experiment is also a lab experiment (not a field experiment): "...a separate N addition [lab] experiment..."

Reviewer #3 (Remarks to the Author):

This is a review for the Nature Communications submission «NCOMMS-22-38305A" with the title "Contrasting geochemical and fungal controls on decomposition of lignin and soil carbon at continental scale" by Huang et al.

I am a reviewer of the original submission and have read the responses to the comments provided by the authors as well as the revised document as a stand along.

I think the authors did a very good job in clarifying the initial comments. The choice of models as well as the statistical approaches and data selection (field vs. lab) are now much clearer. The work is (as the original submission) at a very high level of quality. It is almost a pity that so much material has to move into the supplement, especially the PDP analysis. I really enjoyed reading the revised version and I think this is a timely contribution that should be read across research communities in environmental sciences, thus well suited for Nat Comm. I still disagree with the argument against PCA a bit (I suggested rotated PCA btw) and I also think that the argument that the regression coefficients of PCA approaches is lower is not a strong argument to discard it (The authors make other arguments for discarding it which I agree more with). However, the authors clarified why they chose a different pathway of analyzing this data and there is nothing wrong with that approach. Thus, I don't have many things to add. Below a list of language comments that are optional

l. 93. "lignins" the plural is somewhat unusual even though there is nothing wrong with it. Type of lignin instead?

l.137. instead of depressed: Should this be suppressed or repressed?

l. 316. You could add here "...soil minerals through secondary mineral formation".

l. 377. Highlight or high light?

Figure 1. The information content of figure 1 is a bit low given its size. If the idea is to give an overview on the study setting I would rather say color the map according to soil types or climate zones. I don't find panel b-d very useful at least not in the way it is presented. Why single out the predictors, why show such a simplified version of the sampling and experimental design – The simplification makes it less self-explanatory which I think is the main purpose of such an overview figure. Additionally, the color scheme looks off in panel d between colored fields and text.

Figure 2. Is there a way to increase the clarity of this figure to bring out the main messages presented by this? Right now it looks more like a raw data presentation thing.

Figure 3. Position of "lignin" and "litter" looks a bit lost

RESPONSE TO REVIEWERS' COMMENTS

Reviewer #2 (Remarks to the Author):

Overall, the authors did a great and very thorough job of addressing the reviews. The revised manuscript is very clear and more compelling. Excellent work. One minor comment is that I would suggest adding “Lab” to L99 to make it clear that the N addition experiment is also a lab experiment (not a field experiment): “...a separate N addition [lab] experiment...”

Response: Thank you for the reviewer’s positive comments. As suggested, we have added “lab” to this sentence (L99): “Lignin decomposition is also measured in a separate N addition lab experiment and in field-incubated samples.”

Reviewer #3 (Remarks to the Author):

This is a review for the Nature Communications submission “NCOMMS-22-38305A” with the title “Contrasting geochemical and fungal controls on decomposition of lignin and soil carbon at continental scale” by Huang et al.

I am a reviewer of the original submission and have read the responses to the comments provided by the authors as well as the revised document as a stand along.

I think the authors did a very good job in clarifying the initial comments. The choice of models as well as the statistical approaches and data selection (field vs. lab) are now much clearer. The work is (as the original submission) at a very high level of quality. It is almost a pity that so much material has to move into the supplement, especially the PDP analysis. I really enjoyed reading the revised version and I think this is a timely contribution that should be read across research communities in environmental sciences, thus well suited for Nat Comm. I still disagree with the argument against PCA a bit (I suggested rotated PCA btw) and I also think that the argument that the regression coefficients of PCA approaches is lower is not a strong argument to discard it (The authors make other arguments for discarding it which I agree more with). However, the authors clarified why they chose a different pathway of analyzing this data and there is nothing wrong with that approach. Thus, I don’t have many things to add.

Response: Thank you for reviewing our manuscript once more and the positive comments on our manuscript.

Below a list of language comments that are optional

(1) l. 93. “lignins” the plural is somewhat unusual even though there is nothing wrong with it. Type of lignin instead?

Response: As suggested, we have revised this sentence to (L93–94) “...high-molecular-weight synthetic lignin provides...”

(2) 1.137. instead of depressed: Should this be suppressed or repressed?

Response: We respectfully disagree with the reviewer's comment here. The word "Depressed" here is more appropriate to describe the meaning of driving something low. The word "suppressed" focusses more on the agent that is pushing down, and it implies cause and effect strongly. The word "repressed" would generally be used instead as an adjective and in a political context.

(3) 1. 316. You could add here "...soil minerals through secondary mineral formation".

Response: As suggested, we have added "through secondary mineral formation" at the end of this sentence (L316) "...correlation with soil minerals through secondary mineral formation..."

(4) 1. 377. Highlight or high light?

Response: We have revised this sentence to "this might differ in environments subject to photodegradation" (L378).

(5) Figure 1. The information content of figure 1 is a bit low given its size. If the idea is to give an overview on the study setting I would rather say color the map according to soil types or climate zones. I don't find panel b-d very useful at least not in the way it is presented. Why single out the predictors, why show such a simplified version of the sampling and experimental design – The simplification makes it less self-explanatory which I think is the main purpose of such an overview figure. Additionally, the color scheme looks off in panel d between colored fields and text.

Response: As suggested, we have remade Figure 1. For panel (d), we have simplified the predictors of decomposition by listing the names for the categories of the predictors examined in our study. The color scheme in panel c has also been made consistent between the colored fields and text. We elected to use a simple geographical map without shading by predictors, to avoid misinterpretation and confusion.

(6) Figure 2. Is there a way to increase the clarity of this figure to bring out the main messages presented by this? Right now it looks more like a raw data presentation thing.

Response: As suggested, we now have used the averaged value across the two soil depths for each site to make the plot (Figure 2a, c, e), which reduced the lines and points in these panels but still delivered the message of temporal changes in C decomposition.

(7) Figure 3. Position of "lignin" and "litter" looks a bit lost

Response: As suggested, we have changed the position of the legend for "lignin" and "litter". Please see Figure 3.